

# An Updated, Homogeneous, and Declustered Earthquake Catalog for South Korea and Neighboring Regions

Soumya Kanti Maiti[1], Byungmin Kim[1]*

[1]Department of Civil Urban Earth and Environmental Engineering, Ulsan National Institute of Science and Technology, 50 UNIST-gil, Ulju-gun, Ulsan, 44919, Republic of Korea.

Correspondence to: Byungmin Kim (byungmin.kim@unist.ac.kr)

**Abstract.** The fundamental components for evaluating seismic hazards and forecasting earthquake events in a region include a complete and homogeneous earthquake catalog. Previously, a few studies were performed to combine earthquake databases from various sources to produce a unified earthquake catalog for the Korean Peninsula. To conduct seismic hazard assessments across these regions, this study proposes creating a comprehensive, up-to-date, and unified earthquake catalog for South Korea and its neighboring regions using data from multiple sources. We collected data from the Korea Meteorological Administration (KMA), the International Seismological Centre (ISC), and the Japan Meteorological Agency (JMA). The earthquake database covers the time-period from 1905 to 2023, and the geographical area spans 31°–43° N and 122°–132.5° E. As creating a new earthquake catalog entails combining information from many earthquake record sources, we avoided duplication of occurrences that may arise during the integration process by carefully analyzing the timing and location criteria for each earthquake event. To unify the magnitude scale and produce a homogeneous earthquake catalog, both global and regional empirical equations were used to convert the moment magnitude ($M_W$) and other reported magnitude scales. The resulting homogeneous catalog comprises 63,298 earthquake events, with $M_W$ ranging from 2.0 to 7.9. Declustering of the homogeneous catalog was then conducted to remove dependent events, such as foreshocks and aftershocks, and to identify the mainshocks. Four declustering methods were used to compare and examine their individual influences on mainshock identification in the catalog. The resulting unified and declustered earthquake catalog provides a useful and dependable database for seismicity analysis, seismotectonic studies, and seismic hazard assessments in and around South Korea.

**Keywords:** Homogeneous earthquake catalog, Declustering, Catalog completeness, South Korea.


## 1    Introduction

In the present study, significant research has been conducted to prepare a complete and
unified earthquake catalog for South Korea and its neighboring regions. Although the Korean
Peninsula is not located directly on a tectonic boundary, earthquakes have occurred in the
region since historic times. To gain a thorough understanding of earthquake occurrences in a
specific area, a comprehensive and cohesive earthquake catalog is essential for seismologists,
geologists, policymakers, engineers, and communities because it forms the foundation for
risk assessment, hazard mitigation, and resilient infrastructure development. Every year,
numerous organizations publish earthquake records in the form of bulletins, including the
United States Geological Survey (USGS), the International Seismological Centre (ISC), the
Korea Meteorological Administration (KMA), the Japan Meteorological Agency (JMA), and
the European-Mediterranean Seismological Centre (EMSC). Building on these bulletins,
various researchers worldwide (e.g., Das & Meneses, 2021; Giacomo et al., 2018;
Makropoulos et al., 2012; Rovida et al., 2022; Tan, 2021), systematically develop earthquake
catalogs, which integrate diverse datasets and serve as essential inputs for Probabilistic
Seismic Hazard Assessment (e.g., Anbazhagan et al., 2009; Du & Pan 2020; Tselentis &
Danciu, 2010; Simeonova et al., 2006; Mahmood et al., 2020; Danciu et al., 2024). Studies
on earthquake catalogs in Korea have been conducted over several decades, with significant
contributions from Li (1986), Kim and Gao (1995), and Lee (1999). Since the Korea
Meteorological Administration (KMA) has strengthened its national seismological
observation network, recent efforts have focused primarily on estimating historical
earthquakes (Lee & Yang, 2006; Seo et al., 2010). Seismic hazard studies in Korea typically
use earthquake data from the KMA database (Han & Choi, 2008; Kyung et al., 2016). Ideally,
a comprehensive earthquake catalog should be compiled by integrating earthquake data from
all available sources, not just regional ones. Recent seismic hazard research by Park et al.
(2021) identified this issue and incorporated instrumental earthquake catalogs from the KMA,
JMA, and the China Earthquake Administration (CEA) for their analysis. However, their
database was limited to South Korea, and their primary focus was on seismic hazard studies
rather than catalog details. By contrast, our study aimed to prepare a homogeneous catalog
encompassing the entire Korean Peninsula. In addition, detailed descriptions and an updated
catalog are provided as electronic supplementary material, intended to aid in understanding
seismic activity in the region and to enhance earthquake-related research and preparedness
efforts. Seismic catalogs typically incorporate various magnitude scales, including measures,



such as local magnitude ($M_L$), body-wave magnitude ($M_b$), surface wave magnitude ($M_s$), duration magnitude ($M_d$), velocity magnitude ($M_V$), and moment magnitude ($M_W$). Therefore, converting the various magnitude scales into a unified magnitude scale was necessary. The $M_L$, $M_s$, $M_b$, $M_D$, and $M_V$ magnitude scales exhibit saturation effects at certain levels for significant earthquakes. In addition, these scales display non-uniform behavior across various magnitude ranges. To overcome this limitation, the $M_W$ scale was considered the most reliable, as it directly links the seismic moment to earthquake magnitude, ensuring consistent behavior across all magnitude ranges. Thus, the main objective of this study is to compile a homogeneous moment magnitude ($M_W$) based earthquake catalog for an area comprising South Korea and its neighboring regions. The earthquake database covers the time-period from 1905 to 2023 and the geographical area spans 31° to 42° N and 122° to 132.5° E, with a magnitude range of $M_W$ from 2.0 to 7.9.

Earthquakes are regarded as a complex phenomenon, forming clusters in both space and time, which introduces a bias in seismic catalogs. Consequently, declustering is deemed essential in seismic studies, particularly in probabilistic seismic hazard and regional seismicity analyses (Anbazhagan et al., 2019; Joshi et al., 2023; Taroni & Akinci, 2021). The declustering process in an earthquake catalog involves identifying independent earthquakes (mainshocks) and dependent events (aftershocks and foreshocks) in a dataset. The purpose is not only to eliminate bias but also to disentangle mainshocks from dependent events. Numerous declustering approaches have been proposed, as outlined by Van Stiphout et al. (2012). These methods include deterministic strategies, such as the window-based method (Gardner & Knopoff, 1974; Uhrhammer, 1986), the cluster method linking to spatial interaction zones (Reasenberg, 1985; Savage, 1972), probabilistic approaches, including the stochastic model (Kagan & Jackson, 1991; Zhuang et al., 2002), and the independent stochastic declustering model (Marsan & Lengline, 2010). The resulting declustered catalogs often exhibit notable differences depending on the chosen method. This discrepancy raises concerns, prompting questions about the selection of the optimal declustering algorithm and its impact on seismic hazard assessment. Consequently, this study aimed to quantify and compare the results of various declustering techniques. In this study, we assess four widely used declustering methods: Gardner and Knopoff (1974), Uhrhammer (1986), Reasenberg (1985), and an independent stochastic declustering method (Marsan & Lengline, 2010).

Therefore, the primary contributions of the current study are listed as follows:



- A comprehensive process for building a unified earthquake catalog for South Korea and its neighboring regions is described.

- A newly compiled unified earthquake catalog for the Korean Peninsula and its neighboring regions has been developed. Electronic supplementary material, including a homogeneous earthquake catalog and a declustered earthquake catalog, is also provided.

- A comparison and evaluation of the effects of various declustering algorithms on a

homogeneous earthquake catalog are described.

- Completeness analysis of all declustered earthquake catalogs was performed, which is essential for the seismicity analysis of a region.

Thus, by critically examining the methodologies employed in earthquake catalog compilation, we sought to enhance the reliability and accuracy of seismic information,

ultimately contributing to more robust seismic hazard assessments.

## 2    Methodologies for Catalog Compilation

This section provides an in-depth overview of the methodology used in this study. The work emphasizes the collection of reliable and relevant data with the goal of enhancing the

overall quality of the earthquake catalog. This improvement aims to minimize uncertainties and provide a more robust earthquake dataset by incorporating both regional and global databases, ensuring detailed coverage that encompasses the entire Korean Peninsula. Fig. 1 depicts a flowchart outlining the methodology adopted in the present study, accompanied by concise descriptions of each step.

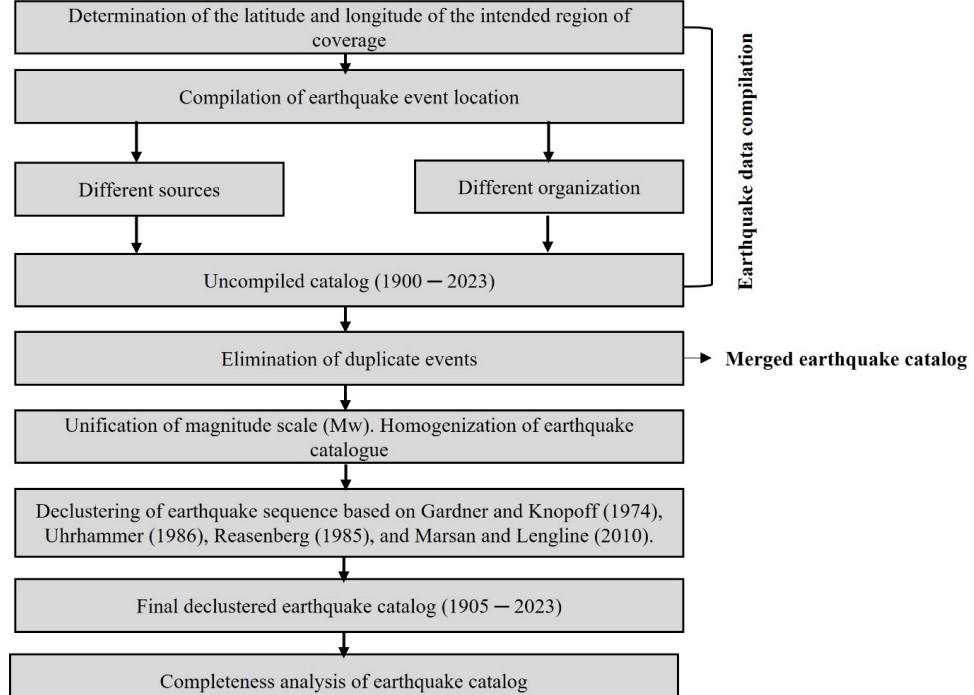


**Figure 1: Flowchart of preparing the earthquake catalog for a region.**

- **Earthquake data compilation:** Initially, historical and instrumental earthquake data were gathered from various agencies, organizations, and global research studies. Raw earthquake data was collected from three global agencies: the Korea Meteorological Administration (KMA), the International Seismological Centre (ISC), and the Japan Meteorological Agency (JMA).

- **Merged earthquake catalog:** In this step, earthquake data collected from various sources was integrated into a single combined dataset. This integration involved a careful examination to identify duplicate events that may exist in the compiled data. Once identified, duplicate events were systematically removed to ensure the integrity and accuracy of the earthquake catalog. The goal was to create a consolidated dataset that avoids redundancy and provides a reliable foundation for subsequent analyses and interpretations in seismic studies.

- **Homogenization of the earthquake catalog:** The standard practice in earthquake catalog studies involves the unification of the magnitude scale by converting commonly reported magnitudes ($M_L$, $M_b$, and $M_S$) into $M_W$. Therefore, in this step, events of all magnitudes were converted into $M_W$.








- **Declustering analysis:** Declustering analysis is the process of removing dependent earthquake events from a homogenized catalog, which is a crucial step in seismicity analysis. In the present study, four declustering algorithms were used to identify mainshocks and aftershocks, which are discussed in detail in subsequent sections.

- **Completeness analysis:** The seismicity of a region varies spatially and temporally. Therefore, statistical analyses using incomplete data may yield unacceptable results. Ensuring the completeness of an earthquake catalog is crucial for seismicity and hazard analyses. In the present study, we employed the methods outlined by Tinti and Mulargia (1985) and Stepp (1972) to conduct the completeness analysis.

### 3.1 Earthquake Data Source and Compilation:

The earthquake data collected for each event in the database included information, such as the date, epicentral coordinates, depth, and earthquake magnitude measured at various scales. To assemble earthquake data for a new earthquake catalog of the Korean Peninsula, we incorporated available data from both national and international seismological databases.

### 3.1.1 Korea Meteorological Administration (KMA) database:

The Korea Meteorological Administration (KMA), the governmental meteorological body of South Korea, is responsible for disseminating information regarding earthquakes and tsunamis. In 1997, the KMA initiated a project to enhance the national seismological observation network and tsunami warning system. Prior to this, there was a lack of adequate earthquake data, necessitating the amalgamation of records from other international agencies. A total of 2,114 events spanning from 1978 to 2023, with magnitudes ranging from 2.0 to 5.8, were collected from the KMA database. All the data included local or regional scale magnitude $M_L$. The seismicity distribution of earthquake locations in the KMA database is shown in Fig. 2.
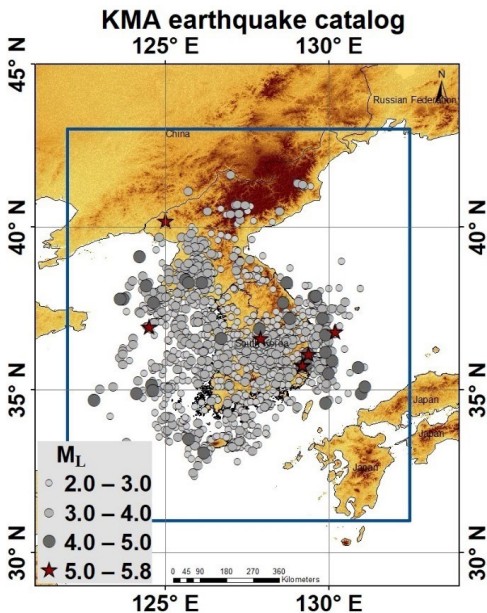

**Figure 2: Seismicity distribution of earthquake locations from the Korea Meteorological**
**Administration (KMA) source.**

### 3.1.2 The ISC bulletin event database:

To produce a new global reference for the earthquake catalog, the International
Seismological Centre's (ISC) bulletin compiled reports on all earthquake data in digital format
starting from 1900. Serving as a comprehensive and refined seismic bulletin, it stands out
internationally when compared with other sources. The bulletin incorporates both raw and
revised earthquake data gathered from approximately 130 local and national networks. The ISC
bulletin expends significant efforts to relocate earthquakes and recalculate their magnitudes,
thereby contributing to the overall reliability of seismic data. For this study, data on 51,894
earthquakes with a magnitude greater than 2.0, covering the time span from 1905 to 2023, were
gathered from the ISC bulletin and documented using various magnitude scales ($M_b$, $M_s$, $M_W$,
$M_L$, and $M_D$). The seismicity distribution of earthquake locations in the ISC database is
depicted in Fig. 3, where M represents all magnitude scales.

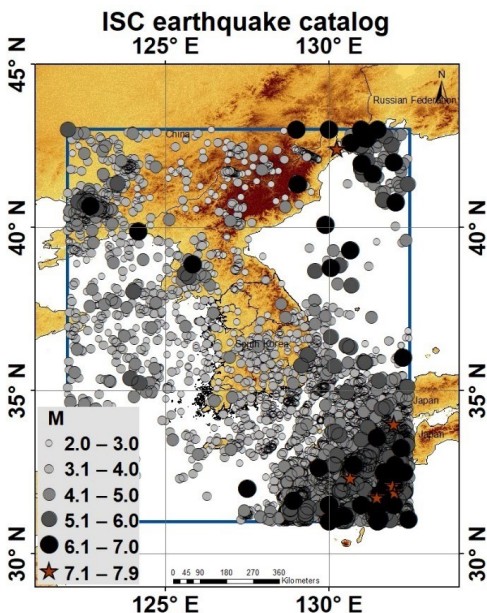

**Figure 3: Seismicity distribution of earthquake locations from the International Seismological Centre (ISC) bulletin. In this figure, M represents various magnitude scales, including $M_b$, $M_s$, $M_w$, $M_L$, and $M_D$.**

### 3.1.3 Japan Meteorological Agency (JMA) database:

The Japan Meteorological Agency (JMA) was the first to make substantial advances in earthquake instrumental measurements and to digitize seismic station bulletin data within and around the Japanese region. This information is regularly updated to create a JMA-unified earthquake catalog in a collaborative effort with the Ministry of Education, Culture, Sports, Science, and Technology (MEXT). Utilizing seismic waveforms from stations affiliated with the JMA, the National Research Institute for Earth Science and Disaster Resilience (NIED), the Japan Agency for Marine Earth Science and Technology (JAMSTEC), universities, and various institutes has contributed to the catalog's comprehensive data. The observed number of seismic events has increased since 2000, primarily due to the implementation of the Hi-net NIED network (Okada et al., 2004). The JMA earthquake catalog includes 48,571 earthquakes spanning from 1919 to 2023, primarily covering the Japanese Islands, south-eastern Korea, and surrounding regions. The magnitudes of the earthquakes in this catalog range from 2.0 to 7.3, as illustrated in Fig. 4.

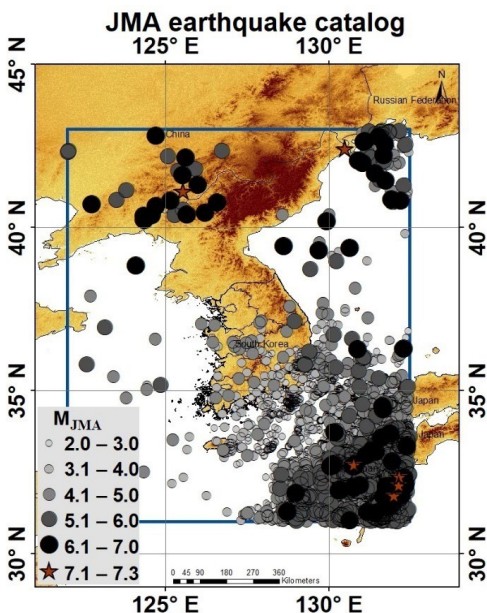

**Figure 4: Seismicity distribution of earthquake locations from the Japan Meteorological Agency (JMA) source.**

### 3.2 Merging of earthquake data from all sources

Data cleaning was required prior to merging earthquake catalogs from the KMA, JMA, and ISC data sources. The unnecessary information, such as the names of source agencies, author names, time zones, region names, and ID numbers, was systematically removed to streamline the datasets. The essential parameters common to all datasets, such as date, time, latitude, longitude, depth, and magnitude, were retained to ensure the structural integrity of the three

data sources. In the subsequent phase, each dataset event was consistently adjusted to the Korean Standard Time (KST) zone to maintain temporal coherence. This step established a unified data structure across all datasets, facilitating seamless merging and manipulation. Finally, a multi-window search technique was applied to remove duplicate events from the catalog. In this multi-window search criteria, the differences in source parameters, such as the

origin time and location of the earthquakes, were used to detect duplicate events. This approach has been used in several studies that combine earthquake catalogs. For example, Mueller (2019) and Petersen et al. (2014) merged multiple regional catalogs of the USGS National Seismic Hazard Model. They used time-window criteria ranging from 10 to 60 s and distance criteria ranging from 20 to 250 km to identify duplicate events. In the Korean region, Park et al. (2021)



used origin time differences of 20 s and distances of 100 km to identify duplicate events across different catalogs. The criteria for these studies were selected based on careful inspection and manual checking of records that correspond to the same event in the compiled catalog. This ensures that the time and distance inputs accurately reflect the characteristics of duplicate events. The window criteria were determined through iterative testing and a manual review of

duplicate events, after which the search window criteria were fine-tuned. A time window of 30 s and a location-distance difference of 70 km were applied to effectively identify and filter duplicate events. In cases where duplicate events were detected, priority was assigned in the order of the KMA, JMA, and ISC, with the highest preference given to events reported by the KMA. Following this cleaning and merging process, the resulting (inhomogeneous) catalog

consisted of 63,298 events with magnitudes ranging from 2.0 to 7.9. This meticulous data cleaning and merging process ensured the creation of a consolidated and reliable earthquake catalog for the comprehensive analysis of the homogeneous earthquake catalog.

## 4 Magnitude homogenization of the earthquake catalog

Numerous researchers worldwide (e.g., Bormann et al., 2007; Bormann & Saul, 2008; Das

et al., 2011; Grünthal et al., 2009; Scordilis, 2006; Sheen et al., 2018; Utsu, 2002) have undertaken the compilation and validation of magnitude scale relations, contributing to the understanding and standardization of seismic measurements. The present study adopted the most globally recognized relations developed by Scordilis (2006) for the conversion of surface wave magnitude ($M_S$) and body wave magnitude ($M_b$) to moment magnitude ($M_W$). The choice

of these relations was rooted in the comprehensiveness and reliability of Scordilis's dataset, which encompasses 20,407 earthquakes sourced from diverse international seismological databases, reflecting seismic events worldwide. The robustness and well-defined nature of the Scordilis (2006) relations make them particularly suitable for accurate magnitude conversions. In the merged earthquake catalog comprising of 63,298 seismic events, the $M_S$ and $M_b$

magnitudes were systematically extracted and subsequently transformed into $M_W$ using the established relations depicted in Eq. (1) and (2). This methodology ensures the consistency and validity of the moment magnitude estimates across a broad spectrum of seismic activities considered in this study.

$$M_W = 0.85(\pm 0.04) * M_b + 1.03(\pm 0.23), \ 2.0 \leq M_b \leq 6.5 \qquad (1)$$






$$M_W = 0.67(\pm 0.005) * M_S + 2.07(\pm 0.03), \ 2.0 \leq M_S \leq 6.1 \hspace{2em} (2)$$

$$M_W = 0.99(\pm 0.02) * M_S + 0.08(\pm 0.13), \hspace{1.5em} 6.2 \leq M_S \leq 8.2$$

The seismic magnitudes recorded by the Japan Meteorological Agency (JMA) seismic network are denoted as $M_{JMA}$ and represent the local magnitude scale, as outlined by Katsumata (2004) and Funasaki et al. (2004). Both the JMA and ISC earthquake catalogs reported events using the $M_{JMA}$ magnitude. Scordilis (2005) provided a calibrated relation to convert from $M_{JMA}$ to moment magnitude ($M_W$) for both strong ($M_{JMA} \geq 5.6$) and weaker ($M_{JMA} \leq 5.5$) seismic events, as expressed in Eq. (3). However, Uchide and Imanishi (2018) identified discrepancies in the magnitude estimations, especially for micro- and small-scale earthquakes. Consequently, they introduced a nonlinear quadratic function, represented by Eq. (4), to enhance the accuracy of $M_{JMA}$ to $M_W$ conversion for these events. In the present study, both relations developed by Scordilis (2005) and Uchide and Imanishi (2018) were used. The final $M_W$ value was estimated by averaging the results obtained from the two equations, providing a more nuanced and comprehensive approach to the magnitude conversion from $M_{JMA}$. In addition, Uchide and Imanishi (2018) highlighted a noteworthy observation regarding the Japan Meteorological Agency (JMA) earthquake magnitude scale. Specifically, they noted that the catalog employs displacement amplitude for larger earthquakes and velocity amplitude for smaller earthquakes to estimate the $M_{JMA}$ magnitude. Consequently, the nonlinear quadratic function presented in Eq. (4) by Uchide and Imanishi (2018) was extended to convert the displacement magnitude ($M_D$) and velocity magnitude ($M_V$) scales to $M_W$. Thus, this equation was incorporated to convert the $M_D$ and $M_V$ magnitudes into $M_W$.

$$M_W = 0.58 * M_{JMA} + 2.25, \ 2.0 \leq M_{JMA} \leq 5.5 \hspace{1.5em} \sigma = 0.28 \hspace{1em} (3)$$

$$M_W = 0.97 * M_{JMA} + 0.04, \ 5.6 \leq M_{JMA} \leq 8.2 \hspace{1.5em} \sigma = 0.22$$

$$M_W = 0.053(\pm 0.003) * M_{JMA}^2 + 0.33(\pm 0.02) * M_{JMA} + 1.68(\pm 0.03) \ \ 0.5 \leq M_{JMA} \leq 7 \ \ (4)$$

In the KMA earthquake catalog dataset, the recorded magnitudes were predominantly on the $M_L$ scale. For a consistent and accurate analysis, a regional relation between the moment magnitude ($M_W$) and local magnitude ($M_L$) was used, based on the work of Sheen et al. (2018). Sheen et al. (2018) conducted a comprehensive study in which they estimated $M_L$ by analyzing both the horizontal and vertical components of seismic events separately. The study utilized 6,327 horizontal and vertical peak amplitudes from 269 earthquakes in the magnitude range of





2.0 to 5.8 that occurred in and around the Korean Peninsula from 2001 to 2016. The vertical peaks and geometrical means of the horizontal peaks were utilized separately to estimate the empirical attenuation curve, station corrections, and earthquake magnitudes accurately. Thereafter, an orthogonal linear regression analysis was performed using the event magnitudes

($M_L$) determined from the horizontal and vertical components along with the $M_W$ values obtained from the S-wave source spectra. It is worth noting that their $M_L$ magnitudes deviated slightly from those derived by the KMA. To match the data, an initial conversion was performed using the relations presented in Eq. (5) and (6), transforming $M_L$ into $M_L^{KMA}$. Afterward, the dataset underwent an additional conversion to obtain the moment magnitude

($M_W$) using the relations outlined in Eq. (7) and (8). The resulting $M_W$ values of these two components were averaged to obtain a consolidated and refined $M_W$ magnitude estimate. This multistep process ensures a unified and standardized magnitude scale for a more accurate and comprehensive seismic analysis.

$$M_L = 0.9187 * M_L^{KMA} + 0.3906 \ for \ horizontal, \tag{5}$$

$$M_L = 0.9234 * M_L^{KMA} + 0.3262 \ for \ vertical, \tag{6}$$

$$M_W = 0.9294 * M_L + 0.3730 \ for \ horizontal, \tag{7}$$

$$M_W = 0.9208 * M_L + 0.4394 \ for \ vertical, \tag{8}$$

Finally, all events were unified using the magnitude conversion equations. The homogenized $M_W$ based earthquake catalog is presented in Fig. 5, encompassing 63,298 events

ranging from 1905 to 2023. A database containing homogenized earthquake events is provided as electronic supplementary material. This additional resource allows researchers and other organizations to access and explore detailed information, enhance transparency, and facilitate further in-depth seismic hazard analyses.



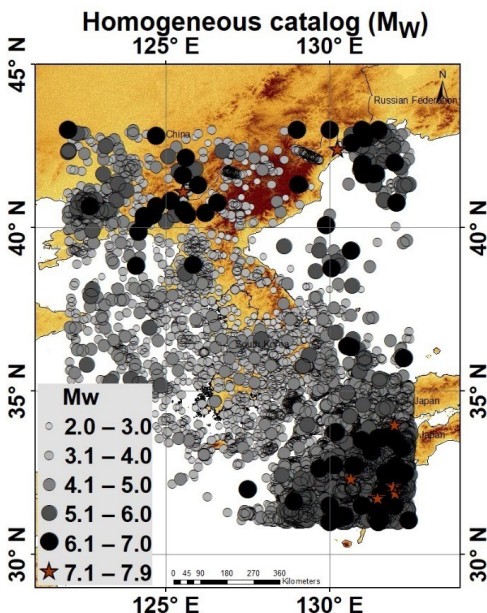

**Figure 5: Homogeneous earthquake catalog for South Korea and neighboring regions covering the period from 1905 to 2023.**

## 5  Declustering of earthquake catalog

The declustering of earthquake catalogs plays a pivotal role in seismic hazard analysis, particularly concerning the fundamental assumptions in Probabilistic Seismic Hazard Assessment (PSHA), in which the earthquake occurrence process adheres to a Poisson distribution. This assumption implies a uniform and random distribution of seismic events, which may be compromised when considering clustered (dependent) events. The presence of clustered events deviates the seismicity from a Poisson distribution by introducing non-random patterns. Moreover, the inclusion of aftershock (dependent event) sequences can lead to an overestimation of the earthquake occurrence rate, potentially resulting in an inaccurate prediction of seismic activity in a region. This necessitates a thorough examination of the declustering process to ensure the reliability of earthquake catalogs, and consequently, to enable more accurate seismic hazard assessments. In this section, we analyzed different declustering algorithms by examining the impact of the adopted processes on a homogeneous earthquake catalog. Various methods have been explored to gauge their effectiveness and their implications for earthquake data analysis. Identifying aftershocks that are dependent on mainshocks poses a challenge because they lack distinct features in their waveforms. Their selection relies on their spatial and temporal proximity to preceding earthquakes or occurrence


rates exceeding the average long-term seismicity. Associating an aftershock with a mainshock necessitates defining a measure for their space-time distance and establishing criteria based on event occurrence. In this study, four types of declustering techniques were employed: window methods, including Gardner and Knopoff (1974) and Uhrhammer (1986); a cluster method utilizing the Reasenberg algorithm (1985); and stochastic declustering implemented through the Marsan and Lengliné approach (2010).

## 5.1 Window-based methods

Windowing techniques offer a straightforward approach for differentiating between mainshocks and dependent events (aftershocks and foreshocks). Each earthquake in the catalog with a magnitude $M_W$ was initially designated as a mainshock. Subsequent shocks were identified as aftershocks if they occurred within a specified time interval T(M) or distance interval L(M). Conversely, the foreshocks were handled in a manner comparable to aftershocks. Specifically, if the most significant earthquake occurred later, preceding foreshocks were reclassified as dependent events. This process involves resetting the time-space windows based on the magnitude of the largest shock in the sequence. Based on the aforementioned assertion, Gardner and Knopoff (1974) provided a mathematical formula for the two determining factors, time and distance, as shown in Eq. (9). Similarly, the time and distance identification criteria provided by Uhrhammer (1986) were estimated using Eq. (10). Table 1 lists the lengths and durations of these windows. The Gardner and Knopoff approach have inspired numerous researchers across generations, with the common goal of distinguishing between background and dependent earthquakes and quantifying the extent of non-randomness in estimated background events.

$$d = 10^{0.1238*M+0.983} \ [km] \qquad t = \begin{cases} 10^{0.032*M+2.7389}, & if\ M \geq 6.5 \\ 10^{0.5409*M-0.547}, & else \end{cases} [days] \qquad (9)$$

$$d = e^{1.77+(0.037+1.02*M} \ [km] \qquad t = e^{-2.87+1.235*M} \ [days] \qquad (10)$$

**Table 1. Time and space windows to eliminate aftershocks.**

|  | Gardner & Knopoff | | Uhrhammer | |
|---|---|---|---|---|
| **M** | **L (km)** | **T (days)** | **L (km)** | **T (days)** |
| **2.5** | 19.61 | 6.39 | 2.68 | 1.24 |
| **3.0** | 22.62 | 11.90 | 4.01 | 2.30 |
| **3.5** | 26.08 | 22.19 | 5.99 | 4.27 |
| **4.0** | 30.07 | 41.36 | 8.95 | 7.92 |





| | | | | |
|---|---|---|---|---|
| **4.5** | 34.68 | 77.10 | 13.38 | 14.69 |
| **5.0** | 39.99 | 143.71 | 20.01 | 27.25 |
| **5.5** | 46.12 | 267.89 | 29.90 | 50.53 |
| **6.0** | 53.19 | 499.34 | 44.70 | 93.69 |
| **6.5** | 61.33 | 884.91 | 66.82 | 173.73 |
| **7.0** | 70.73 | 918.12 | 99.88 | 322.14 |
| **7.5** | 81.56 | 952.58 | 149.31 | 597.35 |
| **8.0** | 94.06 | 988.33 | 223.18 | 1107.65 |

In this study, we utilized the homogeneous earthquake database to implement
declustering method using Eq. (9) and (10): The Gardner and Knopoff algorithm identified
30,912 independent events and 32,386 dependent events, whereas the Uhrhammer algorithm
identified 38,572 independent events and 24,726 dependent events.

The declustered seismicity map distribution of the Korean Peninsula, encompassing the
time span from 1905 to 2023, with the application of the Gardner & Knopoff and Uhrhammer
algorithms, is shown in Fig. 6 and 7, respectively. The maps provide distinct visualizations of
both the mainshock and aftershocks, which are portrayed separately.

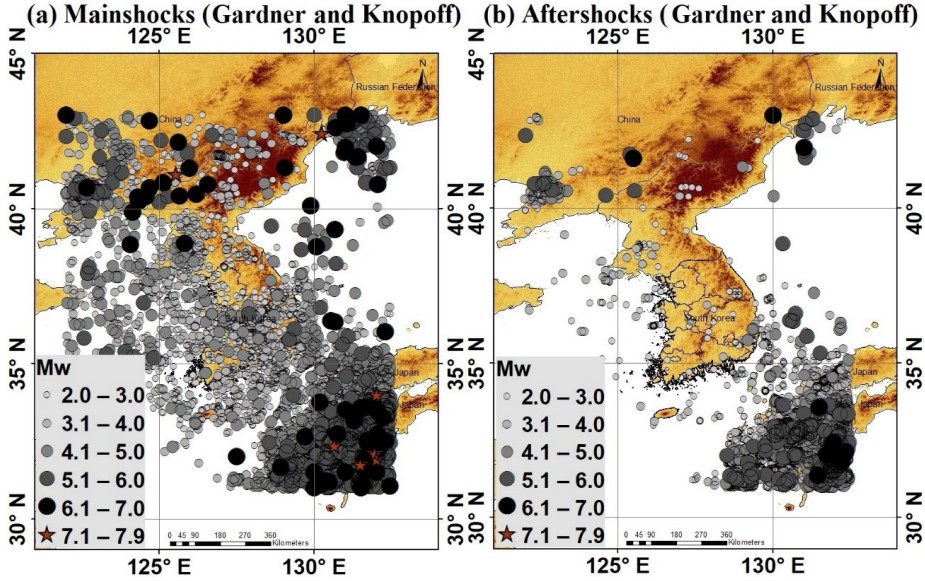

**Figure 6: Declustered seismicity distribution map using the Gardner and Knopoff
method depicting (a) for mainshocks and (b) for aftershocks, covering the period from
1905 to 2023.**


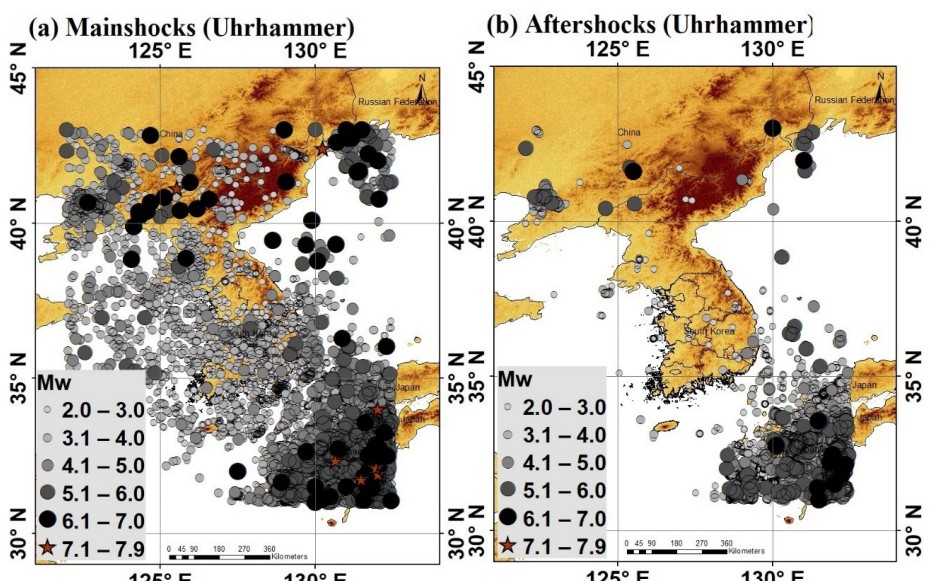

**Figure 7: Declustered seismicity distribution map using the Uhrhammer method depicting (a) for mainshocks and (b) for aftershocks, covering the period from 1905 to 2023.**

Magnitude and time histograms were generated to visualize the distribution of mainshocks and aftershocks, as identified by the Gardner & Knopoff and Uhrhammer algorithms, and are shown in Fig. 8 and 9, respectively.

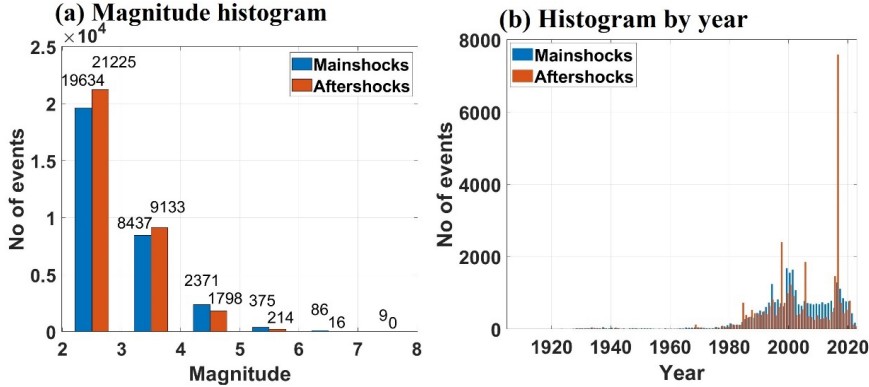

**Figure 8: Histogram plots for (a) number of events by magnitude and (b) temporal distribution of events for mainshocks and aftershocks using the Gardner and Knopoff method.**

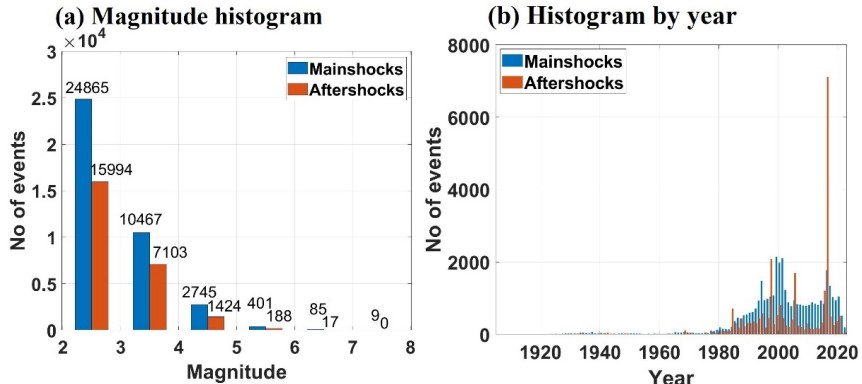

**Figure 9: Histogram plots for (a) number of events by magnitude and (b) temporal**
**distribution of events for mainshocks and aftershocks using the Uhrhammer method.**

## 5.2 Cluster-based Method

Reasenberg (1985) introduced a methodology to identify aftershocks associated with earthquakes by linking events to clusters based on their spatial and temporal interaction zones. Consequently, earthquake clusters tend to expand in size as more earthquakes are included in the analysis. Reasenberg's algorithm establishes a spatial interaction relation defined by the threshold *logd(km) = 0.4M₀ - 1.93+k* (Molchan & Dmitrieva, 1992), where k is one for the distance to the largest earthquake and zero for the distance to the last earthquake. The temporal extension of the interaction zone was determined using Omori's law. All linked events collectively form a cluster, where the largest earthquake is designated as the mainshock, and smaller earthquakes are classified as foreshocks and aftershocks (Van Stiphout et al., 2012). Originally, Reasenberg (1985) focused on identifying foreshocks and aftershocks in central California from 1969 to 1982. Over time, this algorithm has gained popularity in the seismological community. The adoption of standard parameter values from Table 2 has become common practice (Wiemer, 2001; Helmstetter et al., 2006; Mizrahi et al., 2001; Gaudio et al., 2009; Peng et al., 2021; Teng & Baker, 2019). In the algorithm $\tau_{min}$ represents the minimum look-ahead time for constructing clusters when the initial event is not clustered, whereas $\tau_{max}$ denotes the maximum look-ahead time for cluster formation. The parameter $p_1$ signifies the probability of detecting the next clustered event, used in computing the look-ahead time, $\tau$. In addition, $x_k$ represents the increment in the lower cut-off magnitude during clusters: $x_{meff} = x_{meff} + x_k M$, where M is the magnitude of the largest event in the cluster. $x_{meff}$ represents the effective lower magnitude cutoff for the catalog, and $r_{fact}$ signifies the number of crack radii surrounding each earthquake within new events considered to be part of the cluster (Van Stiphout et al.,


2012). For a detailed understanding of these parameters, please refer to the original publication by Reasenberg (1985).

In this study, we utilized a homogeneous earthquake database to implement a declustering method using the Reasenberg algorithm. The default parameters listed in Table 2 were implemented to ensure consistency and reliability of the analysis. By applying the Reasenberg algorithm, 39,978 events were identified as mainshocks and characterized as independent events. In addition, 23,320 events were recognized as aftershocks and foreshocks,

representing dependent events in the seismic sequence.

**Table 2. Input parameters for declustering algorithm by Reasenberg.**

| Parameter | Standard value | Min. value | Max. value |
|---|---|---|---|
| $\tau_{min}$ [days] | 1 | 0.5 | 2.5 |
| $\tau_{max}$ [days] | 10 | 3 | 15 |
| $p_1$ | 0.95 | 0.9 | 0.99 |
| $x_k$ | 0.5 | 0 | 1 |
| $x_{meff}$ | 1.5 | 1.6 | 1.8 |
| $r_{fact}$ | 10 | 5 | 20 |

      A comprehensive declustered seismicity map encompassing the time span from 1905 to 2023, with the application of the Reasenberg algorithm, is presented in Fig. 10. The map

provides distinct visualizations for both mainshocks and aftershocks, portrayed separately in subfigures (a) and (b), respectively. These detailed seismicity maps contribute to a spatial understanding of earthquake distribution, highlighting regions with heightened seismic activity and illustrating the prevalence of aftershocks in the aftermath of mainshock events. The cluster linkage criteria showed a higher density in the active regions of Japan and the southeastern part

of South Korea, suggesting a concentration of dependent seismic events in these areas. Conversely, the density was diffused in other regions, indicating a comparatively lower frequency of clustered events. Such graphical representations enhance the interpretation of seismic patterns and aid researchers and seismologists in discerning the geographical dynamics of earthquake occurrences over a specified period. Magnitude and time histograms were

generated to visualize the distribution of the mainshocks, and aftershocks identified by the Reasenberg algorithm, as shown in Fig. 11. These plots offer a detailed exploration of seismic activity, portraying the frequency and temporal occurrence patterns of both mainshocks and their associated aftershocks. Analysis of these histograms provides a comprehensive overview of the seismic behavior in the studied regions, aiding in the characterization of earthquake

sequences and their temporal evolution.

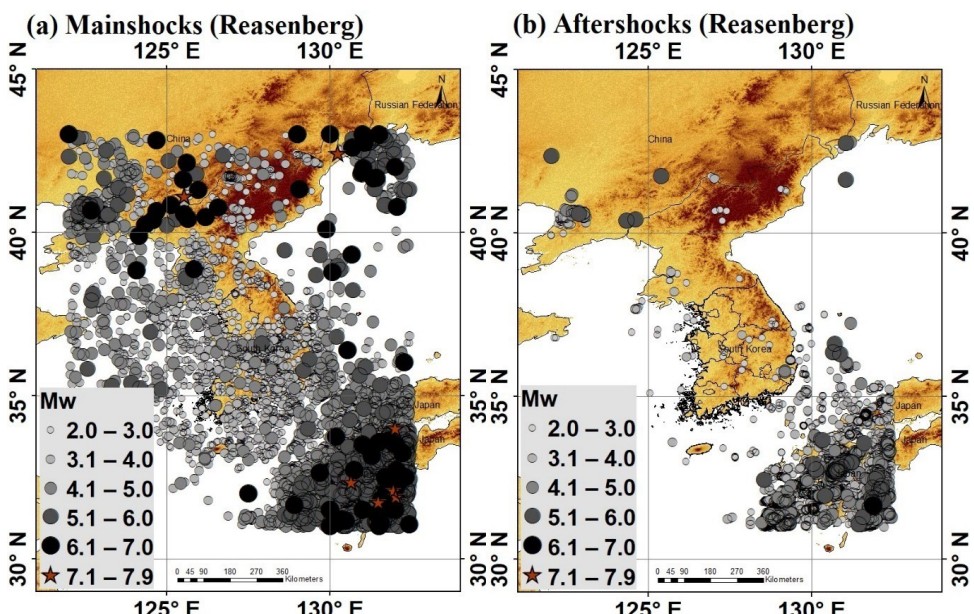

**Figure 10: Declustered seismicity map using the Reasenberg algorithm depicting (a) for mainshocks and (b) for aftershocks, covering the period from 1905 to 2023.**

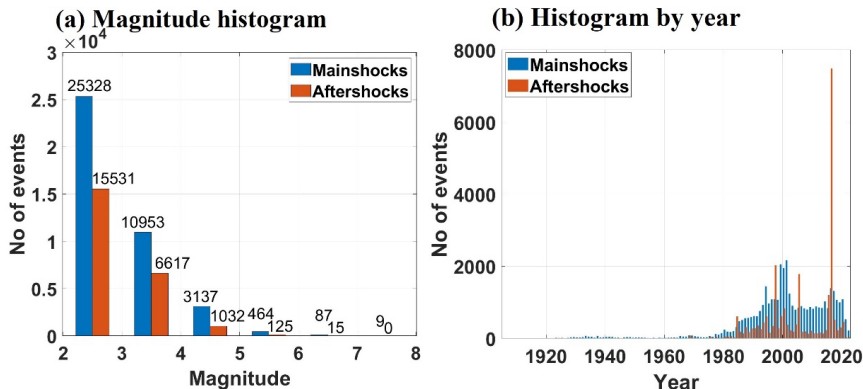

**Figure 11: Histogram plots for (a) number of events by magnitude and (b) temporal distribution of events for mainshocks and aftershocks using the Reasenberg algorithm.**

## 5.3    Stochastic decluster method

The decluster algorithm, which uses window- and cluster-link-based approaches, involves the use of subjectively chosen parameters, such as the size of windows and the distance between linked nodes. Variations in parameter values lead to variations in declustered catalogs and the assessment of background seismicity estimates. Typically, researchers determine these parameters based on prior expertise, using specific datasets. Depending on the declustering



results, a trial-and-error process is often used, with particular attention paid to the temporal smoothness of the resulting catalog. In contrast to deterministic declustering methods and integrating probabilistic treatments into the clustering model, Marsan and Lengliné (2010) proposed the Model-Independent Stochastic Declustering (MISD) approach. This method does not depend on a specific model or parametrization. Essentially, it relies on the fact that seismicity dynamics are a linear cascade of earthquake triggers, which helps to distinguish aftershocks directly triggered by earthquakes from those indirectly triggered (Hainzl & Marsan, 2008). In this method, mainshocks that were sufficiently isolated from other significant seismic events were initially selected to avoid confusion with unrelated earthquakes. Thereafter, the shortest distances from the mainshock to subsequent earthquakes were calculated, providing a more accurate representation of the aftershock. The core of the method involves estimating the probability 'ωAB' that an earthquake 'B' is an aftershock of earthquake 'A', allowing for a continuous range of probabilities, unlike traditional declustering methods where 'ωAB' can only be zero (not an aftershock) or one (aftershock). This probability is derived using stochastic and inversion techniques. The background seismicity was accounted for by comparing the observed aftershock distribution with the expected distribution from normal seismic activity. Parameter optimization, which is achieved by solving nonlinear equations using an expectation-maximization algorithm, ensures accurate and reliable results. This comprehensive approach, which includes modelling both direct and indirect aftershocks and employing the Monte Carlo method for probability estimation, provides a detailed and refined understanding of aftershock patterns and their spatial decay relative to the mainshock. Ideally, this model should be less sensitive to arbitrary parameterization than other declustering methods, thereby enhancing its reliability.

Model-independent stochastic declustering (MISD) analysis was performed using C-based programming code developed by David Marsan. The ISTerre website provides this code, which facilitated the identification of the mainshocks and aftershocks in our study. The utilization of Marsan's code provides a reliable and efficient means for conducting MISD analysis, contributing to the accuracy of our seismic event categorization. From a homogeneous dataset of 63,298 events, the MISD algorithm identified 25,229 mainshock events, of which 38,069 were identified as aftershocks. Notably, this method is more sensitive to seismic patterns than the Reasenberg algorithm and produces a larger number of aftershocks. Fig. 12 depicts the declustered seismicity map, spanning 1905–2023, employing the MISD algorithm to categorize the event distribution of the (a) mainshocks and (b) aftershocks. Furthermore, Fig.




13 enriches our analysis by showing the magnitude and time histogram plots, providing insights into the distribution and temporal occurrence patterns of both mainshocks and aftershocks in terms of both magnitude and time.

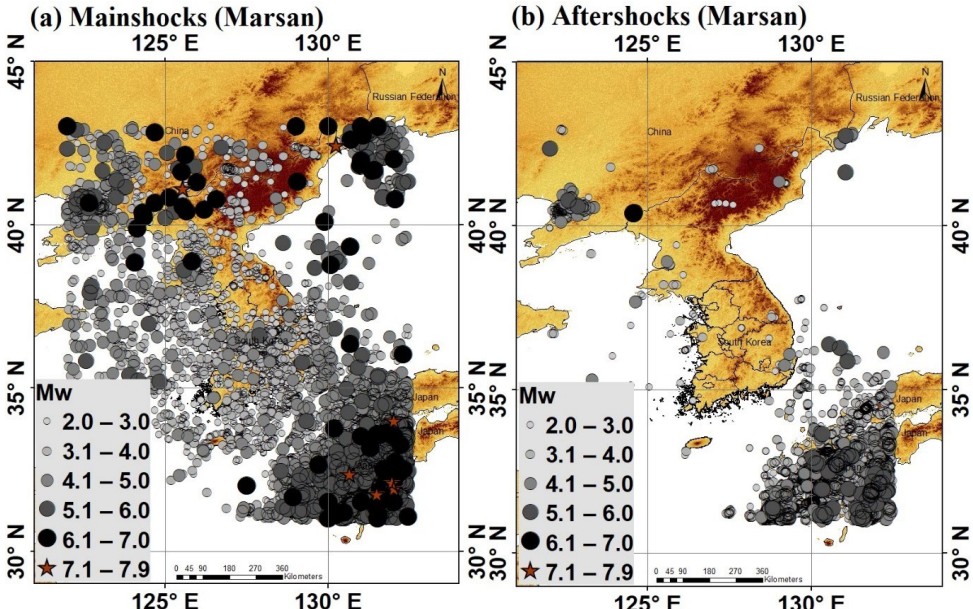

**Figure 12: Declustered seismicity map using the Marsan algorithm depicting (a) for mainshocks and (b) for aftershocks, covering the period from 1905 to 2023.**

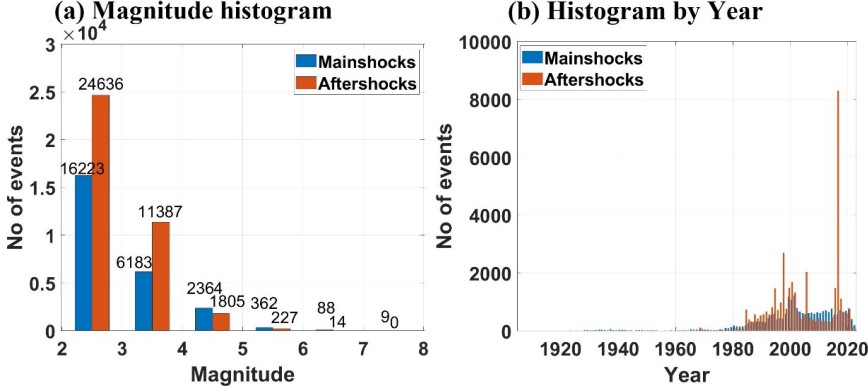

**Figure 13: Histogram plots for (a) number of events by magnitude and (b) temporal distribution of events for mainshocks and aftershocks using the Marsan algorithm.**


## 6 Completeness analysis:

The earthquake catalogs exhibit spatial and temporal irregularities. These inconsistencies arise from variations in spatial coverage, changes in network configuration over time, and advancements in the constituent instruments. Hence, an earthquake catalog must be complete with respect to the relative frequency of earthquake occurrences over time. As earthquake catalogs are inherently incomplete across the magnitude range covered, a thorough completeness analysis was conducted. This analysis established the magnitude threshold ($M_c$) above which all events were reliably recorded. The estimation of $M_c$ is essential for various seismological statistical analyses and plays a critical role in estimating the parameters of the Gutenberg–Richter (GR) law, namely, the a- and b-values. The seismicity recurrence parameters may exhibit bias without accurate completeness analysis, leading to inappropriate estimations in seismicity analysis and probabilistic seismic hazard assessments (Bayliss & Burton, 2007; Popandopoulos et al., 2016; Sawires et al., 2019). In general, earthquake catalogs become sparser and more uncertain when observed backward in time, indicating that completeness periods fluctuate over time. For large earthquakes, the completeness period extends to pre-instrumental or historical times. Conversely, for small-magnitude earthquakes, completeness was achieved only in the most recent decades of the instrumental epoch because instrumental recording was not available in the past, resulting in the non-recording of many smaller-magnitude events in the region.

The completeness periods and threshold magnitudes were estimated individually for the four sets of declustered catalogs. In this study, we focused on determining completeness analysis using two methods. The first is the cumulative visual inspection (CUVI) method proposed by Tinti and Mulargia (1985), and the second is based on statistical analysis by Stepp (1972). In the CUVI method, a graph plotting the cumulative number of earthquakes against time duration was generated. The catalog was deemed complete during periods when the earthquake occurrence rate remained constant. It is conventionally assumed that the latest change in slope signifies when the data are complete, and the interval with the highest slope is chosen. In this section analysis conducted using the Gardner and Knopoff declustered earthquake catalog is presented. The results stemming from this particular catalog were discussed in the main portion of the text. For a comprehensive view, the results obtained from the utilization of alternative declustered catalogs are provided in the electronic supplementary material. This separation ensures clarity and allows readers to explore the outcomes of the



various declustered datasets without complicating the primary analysis presented in the main
text. In this study, a completeness analysis was conducted by dividing the declustered
earthquake catalog into magnitude intervals, starting from a magnitude of 2.0 with an increment
of one. The cumulative number of events for each magnitude bin was calculated and plotted
against time-period as shown in Fig. 14. A graph depicting the cumulative number versus time-
period was visually inspected to identify the point at which the curve becomes a straight line.
This point was considered the period of completeness for the magnitude bins, and the calculated
completeness period for each magnitude is tabulated in Table 3.

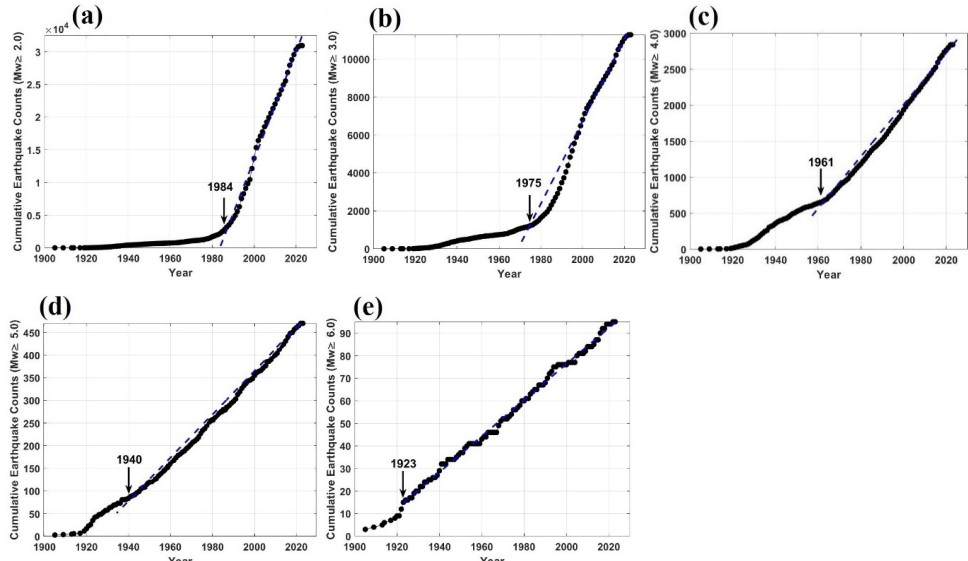

**Figure 14: Completeness analysis for the Gardner and Knopoff declustered catalog using**
**the CUVI method for each magnitude range: (a) Mw ≥ 2.0 (b) Mw ≥ 3.0 (c) Mw ≥ 4.0 (d)**
**Mw ≥ 5.0 (e) Mw ≥ 6.0. Arrows indicate the completeness year.**

The second method, proposed by Stepp (1972), relies on the assumption that earthquake
occurrences within each magnitude subclass adhere to a Poisson distribution when represented
as a point process over time. In this regard, the catalog was segmented into five magnitude bins:
$2 \leq M_W < 3$, $3 \leq M_W < 4$, $4 \leq M_W < 5$, $5 \leq M_W < 6$, $M_W \geq 6$. Subsequently, the average number of
earthquakes per decade was estimated for each magnitude bin. Let the number of events per
unit time interval be defined as $x_1, x_2, \ldots, x_n$ for each magnitude bin; the unbiased estimate of
the mean rate per unit time interval is given by Eq. (11):

$$\lambda = \frac{1}{n} \sum_{i=1}^{n} x_i \qquad (11)$$



where n denotes the unit time interval. The variance is inversely proportional to the length of the sample period and is defined as $\sigma_\lambda^2 = \frac{\lambda}{T}$, where 'T' is the duration of the sample. The standard deviation of the mean occurrence is defined as $\sigma_\lambda = \sqrt{\frac{\lambda}{T}}$. A plot of Stepp's method for the five magnitude bins is shown in Fig. 15, where the standard deviation of the mean rate for

the different magnitude bins versus time is plotted. Thereafter, tangent lines with a slope of $1/\sqrt{T}$ for each magnitude bins are plotted. The completeness period of that magnitude class is identified during the period when the data followed a trend parallel to the tangent line. The point at which this downward trend deviates indicates the beginning of an incomplete period of catalog reporting for a specific magnitude interval.

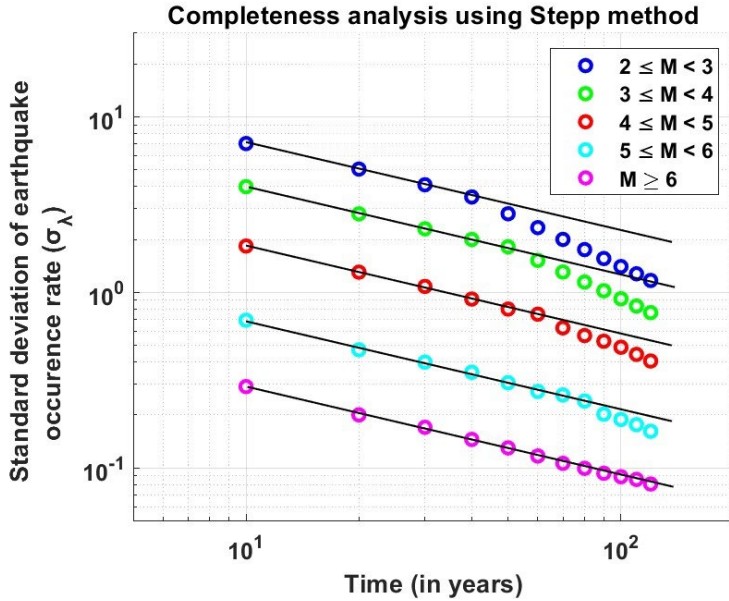


**Figure 15: Completeness analysis of the Gardner and Knopoff declustered catalog for different magnitude ranges using Stepp's method.**

      A summary of the completeness analysis using the two methods (CUVI and Stepp) for all four declustered catalogs is presented in Table 3 for comparison. In the past, only

significantly large earthquake events were recorded or reported. However, smaller earthquakes began to be recorded as the seismograph network expanded and became more sensitive over time. The completeness results based on both methods indicate the completeness level of small-to moderate-magnitude earthquakes has acheived over the last 40 years with advancements in seismic monitoring. Interestingly, the completeness period tended to increase





555 for higher-magnitude earthquakes, showcasing the limited capabilities of the seismograph network to capture moderate-to-large seismic events across a broader range over time. The completeness analysis results for both methods yielded consistent ranges of completeness periods (Table 3). The similarities between the findings of the two methods suggest a degree of validation and mutual support, reinforcing the accuracy of the identified completeness

560 periods for seismic events. Electronic supplementary information includes detailed information on the periods of magnitude completeness obtained through other declustered catalogs. The completeness analysis results for the multiple declustering catalogs enhanced the comprehensiveness of the study, allowing for a thorough comparison of their findings. Notably, the completeness period for the magnitudes across these declustered catalogs demonstrated

565 similar results, reflecting a consistent trend in earthquake recordings over time. Despite variations in the mainshocks identified by each declustered catalog, the coherence in the completeness periods suggests a shared understanding of seismicity patterns within the studied region.








**Table 3. Completeness periods for different magnitude classes using the CUVI and Stepp methods.**

| Magnitude | Gardner Method | | | | Uhrhammer Method | | | | Reasenberg Method | | | | Marsan Method | | | |
|---|---|---|---|---|---|---|---|---|---|---|---|---|---|---|---|---|
| | CUVI method | | Stepp Method | | CUVI method | | Stepp Method | | CUVI method | | Stepp Method | | CUVI method | | Stepp Method | |
| | Period | Interval | Period | Interval | Period | Interval | Period | Interval | Period | Interval | Period | Interval | Period | Interval | Period | Interval |
| Mw ≥ 2.0 | 1984 - 2023 | 37 years | 1983 - 2023 | 40 years | 1985 - 2023 | 38 years | 1983 - 2023 | 40 years | 1986 - 2023 | 37 years | 1983 - 2023 | 40 years | 1982 - 2023 | 41 years | 1983 - 2023 | 40 years |
| Mw ≥ 3.0 | 1975 - 2023 | 47 years | 1973 - 2023 | 50 years | 1976 - 2023 | 47 years | 1973 - 2023 | 50 years | 1978 - 2023 | 45 years | 1973 - 2023 | 50 years | 1975 - 2023 | 48 years | 1973 - 2023 | 50 years |
| Mw ≥ 4.0 | 1961 - 2023 | 59 years | 1963 - 2023 | 60 years | 1964 - 2023 | 59 years | 1963 - 2023 | 60 years | 1964 - 2023 | 59 years | 1963 - 2023 | 60 years | 1960 - 2023 | 63 years | 1963 - 2023 | 60 years |
| Mw ≥ 5.0 | 1940 - 2023 | 83 years | 1943 - 2023 | 80 years | 1940 - 2023 | 83 years | 1943 - 2023 | 80 years | 1940 - 2023 | 83 years | 1943 - 2023 | 80 years | 1939 - 2023 | 84 years | 1933 - 2023 | 90 years |
| Mw ≥ 6.0 | 1923 - 2023 | 100 years | 1923 - 2023 | 100 years | 1924 - 2023 | 99 years | 1923 - 2023 | 100 years | 1924 - 2023 | 99 years | 1923 - 2023 | 100 years | 1923 - 2023 | 100 years | 1923 - 2023 | 100 years |









## 7 Results and discussion

The results of the earthquake declustering analysis conducted for the Korean Peninsula using four discrete methods — Gardner and Knopoff, Uhrhammer, Reasenberg, and Marsan and Lengliné reveal variations in the categorization of aftershocks and dependent events, as presented in Table 4. Different methods have distinct criteria for identifying dependent events. Consequently, these procedures yield different numbers of independent and dependent events. In particular, the Gardner and Knopoff algorithm exhibited a notable tendency to classify a significant proportion (51%) of events with stronger shaking as dependent events, characterizing them as aftershocks. Notably, a key contributing factor to the higher count of dependent events observed with the Gardner and Knopoff algorithm was the large spatial window employed, which was originally developed based on the seismic characteristics observed in California earthquakes. By contrast, the Uhrhammer method, while identifying dependent events, uses a more conservative approach, resulting in a lower percentage of aftershocks. The Reasenberg algorithm demonstrated the most conservative stance, removing the minimal number of stronger dependent events and accounting for approximately 37% of the datasets as aftershocks. Importantly, the Marsan and Lengliné method introduces a new perspective on the stochastic approach and classifies fewer mainshocks, accounting for only 40% of the dataset. This method displays a distinctive tendency to identify a higher proportion of aftershocks, contributing to a more restrained count of mainshocks. This observation adds a layer of complexity to the overall findings, suggesting that the algorithm tends to overestimate aftershocks compared to other traditional methods.

**Table 4. Summary of results of declustering techniques for the Korean region.**

| Method | Mainshocks | Aftershocks | Percentage |
|---|---|---|---|
| Gardner Method | 30912 | 32386 | M = 49%, A = 51% |
| Uhrhammer Method | 38572 | 24726 | M = 61%, A = 39% |
| Reasenberg Method | 39978 | 23320 | M = 63%, A = 37% |
| Marsan Method | 25229 | 38069 | M = 40%  A = 60% |

The time-series analysis of the cumulative seismicity for each algorithm is visually depicted in Fig. 16 for the mainshocks and Fig. 17 for the aftershocks. These plots provide a comprehensive representation of the cumulative number of seismic events over time. In Fig. 16, which focuses on the mainshocks, the trends observed across the algorithms provide valuable insights into the temporal distribution of significant seismic events in the region. Each algorithm's distinctive declustering approach becomes apparent and influences the





accumulation of mainshocks over time. Notably, the Reasenberg algorithm, with its tendency to classify a higher proportion of events as mainshocks, may exhibit a steeper upward trajectory in the cumulative seismicity plot compared with the more conservative approaches of the others. By contrast, in Fig. 17, the Marsan and Lengliné method, with its tendency to identify more aftershocks, may show a steeper pattern in the cumulative plot. Time-series plots serve as tools

to assess the performance of declustering algorithms over time, allowing researchers and seismic hazard practitioners to identify patterns, anomalies, and potential areas of improvement. In addition, the plots for mainshocks and aftershocks provide valuable information for refining declustering methodologies and enhancing our understanding of seismic activity in the Korean Peninsula. For a visual representation, the time of occurrence vs. Latitude-Longitude

comparison plot of the mainshocks obtained from the four declustering algorithms is depicted in Fig. 18. The seismic activity pattern in the region indicates that the maximum earthquake concentration occurred in a magnitude ($M_W$) range of two to five. This implies that the region predominantly experienced low-to-moderate seismic activity. The earthquake cluster indicates a relatively stable seismic environment for the Korean Peninsula, with occasional moderately

high tremors. Although the occurrence of large seismic activity cannot be entirely dismissed, the predominance of earthquakes in the magnitude range of $M_W$ two to five suggests a lower likelihood of destructive events. It is essential to recognize that this conclusion is based primarily on earthquake data from the last 40 years, a period marked by significant advancements in seismic monitoring technology. Consequently, the data from this era are more

comprehensive and accurate. However, earthquake datasets prior to this period were less complete, potentially underestimating the frequency and magnitude of historical seismic events. Therefore, although recent data indicate a reduced likelihood of significant seismic events, the limitations of earlier data introduce a degree of uncertainty in long-term seismic hazard assessments.


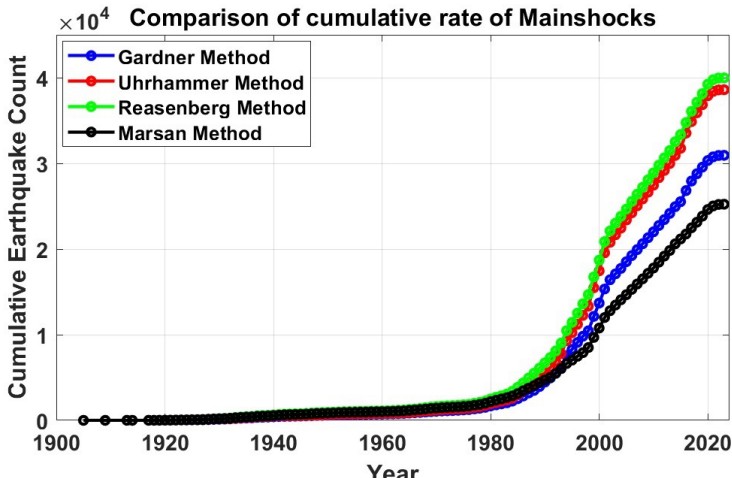


**Figure 16: Time-series plot of cumulative seismicity of mainshocks for each algorithm.**

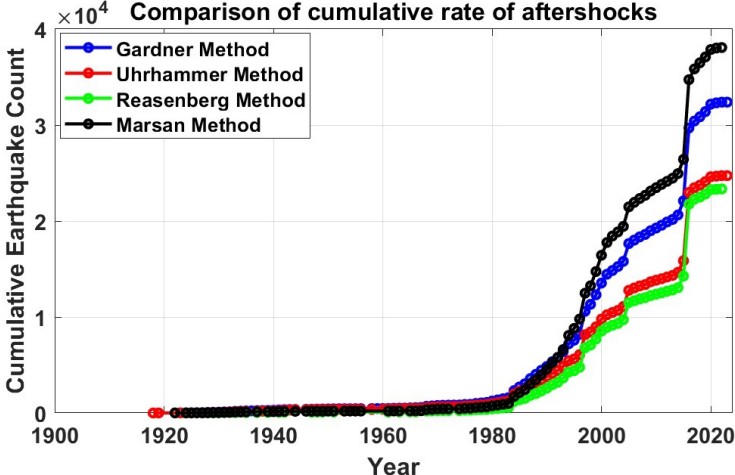

**Figure 17: Time-series plot of cumulative seismicity of aftershocks for each algorithm.**

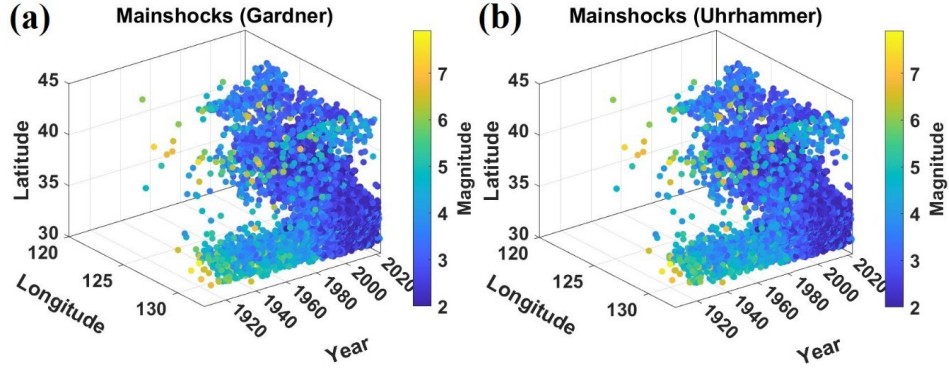



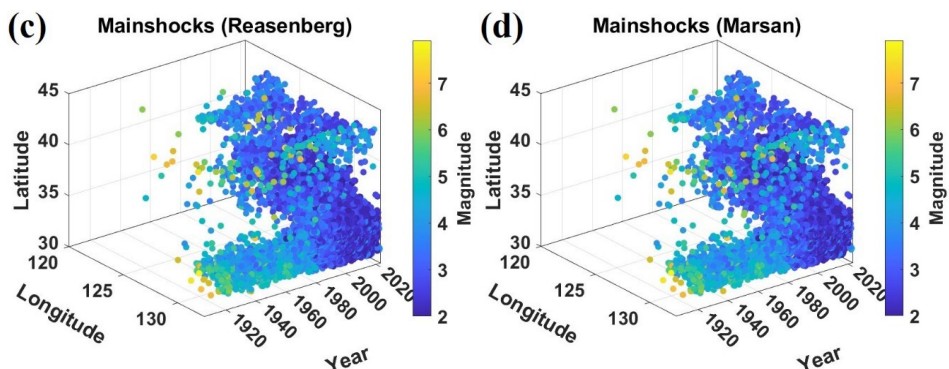


**Figure 18: Time of Occurrence vs. Latitude-Longitude Comparison plot of Mainshock earthquakes using (a) Gardner method (b) Uhrhammer method (c) Reasenberg approach (d) Marsan method.**

## 8 Conclusions

In this study, we prepared a homogeneous and complete earthquake catalog for the Korean Peninsula and observed disparities in the declustering results. In South Korea, there is a high probability of seismic events in Gyeongsangnam-do Province (south-eastern region), highlighting the importance of seismic hazard evaluation for the region. The present study emphasizes the importance of evaluating the effectiveness of declustering techniques in seismic

catalogs, focusing not only on the temporal properties of events but also on their spatial features through clustering techniques. Additional electronic supplementary material is available, including a homogenized earthquake catalog, a declustered earthquake catalog, and the results of the completeness analysis. Researchers can access these resources in the supplementary Materials section. A particular approach may not be suitable for different tectonic settings,

emphasizing the need for continued refinement and adaptation of declustering algorithms to enhance their accuracy in seismic hazard assessments. This study provides valuable insights into seismic activities in the South Korean region and serves as a foundation for further research to optimize declustering methodologies for enhanced seismic risk evaluation and mitigation strategies. Nonetheless, identifying the most effective method for removing dependent

earthquakes is challenging because there is no inherently unique approach, and the elimination results are not absolute. A future extension of this work involves estimating the seismic hazards with a specific focus on discerning the distinct effects associated with each declustered catalog. A detailed PSHA is necessary to confirm the possible influence on the uniform hazard spectra and disaggregation analysis. This is important because different methods of studying

earthquakes may yield different results. By comparing these results, we can improve our
understanding of earthquake risks associated with the declustering method and develop better
plans to keep people and structures safe. This homogeneous and declustered earthquake catalog
will serve as a reliable source for evaluating seismicity parameters and seismic hazards in
Korea and its surrounding regions.

**Data Availability Statement**

The earthquake data were downloaded from the websites: http://necis.kma.go.kr/ for KMA,
https://www.isc.ac.uk/iscbulletin/search/catalogue/ for ISC, and
http://www.data.jma.go.jp/svd/eqev/data/bulletin/index_e.html for JMA (last accessed
December 2023). The homogeneous earthquake catalog and all the declustered earthquake

catalogs are provided in the supplemental materials (.xlsx files). Please refer to the resources
provided and the supplemental materials for further details.

**Supplementary Information**

The supplementary documents include:

1. Homogeneous earthquake catalog (.xlsx format).
2. Declustered earthquake catalog based on four methods (.xlsx format).
3. Completeness analysis results (.docx format).

**Acknowledgements**

The authors are grateful to the National Research Foundation of Korea (NRF) and Brain Pool
Fellowship for their support.

**Funding**

This work was supported by the National Research Foundation of Korea (NRF) grant for Brain
Pool Projects (Grant No:2022H1D3A2A02093553).

**Author contributions**

All authors contributed to the conceptualization and execution of the research, analysis of the
results, writing of the manuscript, and supervision of the study.

**Competing interests**

The authors declare that they have no competing interests.




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
