# Peer review of "An Updated, Homogeneous, and Declustered Earthquake Catalog for South Korea and Neighboring Regions"

_Natural Hazards and Earth System Sciences, 2024_

## Referee Comment (RC2)

**An Updated, Homogeneous, and Declustered Earthquake Catalog for South Korea and Neighboring Regions, *by Soumya Kanti Maiti and Byungmin Kim**

The paper presents the compilation of a unified earthquake catalog, spanning 1905 to 2023, as well as a declustering analysis using four different methods and a completeness analysis of the compiled catalog.

The paper is very well written and well presented. I have one significant comment and a few minor suggestions –

I will start from the end – in line 696 you write: "This study provides valuable insights into seismic activities in the South Korean region and serves as a foundation for further research to optimize declustering methodologies for enhanced seismic risk evaluation and mitigation strategies." However, I feel that the insights from the work are very limited. What you present is a catalog compilation and a simple comparison between four methods of declustering. Relatively simple and not very inspiring. While you claim that "identifying the most effective method for removing dependent earthquakes is challenging because there is no inherently unique approach, and the elimination results are not absolute." – At the end – when you (or someone else) goes to estimate seismic hazard – you will have to choose one of the catalogs. Which one would you choose and why ? Will you run with all four ? Simply running a comparison is not enough. It has been done before. You don't have to show a full PSHA here, but can you at least estimate $a$ and $b$ for a couple of different regions with your four declustering methods and comment about the significance of the difference? Maybe it doesn't make a difference after all? I think this type of analysis can take your paper one step forward and help determine whether or not the differences in the clustering approaches even matter.

Minor comments:

1. Section 2- what are the boundaries of your region of interest (please state those explicitly, the reader does not need to infer them from figures). What are the distances between those boundaries to locations of interest within the peninsula ? In other words- show you are not cutting your region too small compared to the locations you want to compute seismic hazard at. For example- JeJu-Do is about 250 from the nearest boundary, Is that sufficient?
2. Line 221 – the criteria for duplicate events is 70km and 30sec. This is quite arbitrary. What about consistency in magnitude estimations? Is that in the criteria? If you have a M2.0 and a M6.0 occuring at the same time within 70km, are those duplicate?
3. Line 255 – When you average magnitude estimations obtained using different approaches – please provide also the Standard deviation of the magnitude estimate. Not only the average. You can add that to your catalog.
4. Line 265 – It isn't clear which euqation is used – 3 or 4? If only 4 – please do not present Eq3. If both – then what is the criteria ?
5. Line 282- The magnitude of an event cannot be different for vertical and horizontal components. Its a source parameter. It was derived in the original paper, but does not make sense here. If you proceed with the averaging- at least present Std of Mw estimate.
6. Line 345 – I think there is a typo in the distance equation. Why would 1.77 be outside the brackets and 0.037 inside if both are constants?

7. Line 438 – "with particular attention paid to the temporal smoothness of the resulting catalog" – what does that mean? what is temporal smoothness ?

8. Line 466 – I think there is a typo in this sentence in using "of which". This reads as if you identified 25,229 mainshocks, and 38,069 out of the 25,229 were identified as aftershocks. This does not make sense.

9. Line 511 – This is still the main portion of the text. Give section number instead.

10. Line 540 – 'bins versus time' – which time? What is 10, and 100? Is that "time before present" ? If so – then it should say so in the text and figure caption.

11. Section 7 –
    The comparison results from both declustering and completeness analysis should be compared with similar studies in the literature.
    e.g .
    DOI: 10.1007/s10950-024-10221-8
    DOI: 10.1785/0220210127.
    And others..

12. Lines 660 until end of paragraph- There is nothing new stated in this paragraph. There are always more small to moderate earthquakes than large ones. Everywhere. The G-R model shows that, as do others. If you only have reliable data from the past 40 years you probably need models to help you estimate the large events. The text here is trivial.

13. Figure 18 – It is quite hard to gain anything from these figures, and I do not believe you say anything significant about them in the text. Are they really necessary ? What can you say about the geographical distribution that can only be understood from these plots (and not the maps for example) ?

---

## Author Response (AR1)

**Response to reviewers to the manuscript numbered NHESS-2024-197**

We would like to express our sincere gratitude to the editor and the reviewers for the thoughtful and constructive comments on our manuscript. We greatly appreciate the time and effort invested in reviewing our work. The suggestions have been highly valuable in improving the clarity and quality of the manuscript. We have carefully addressed all the comments and provided a point-by-point response to each of the three reviewers.

A detailed response to each comment is provided below:

**Response to Reviewers' Comments #1 (RC1)**

**Comment Reviewer #1:**

The study is very well articulated and concise, it provides valuable information and methodologies for assessing the seismic hazard and risk, having in view the seismic activity in the South Korean region and is neighboring. There are presented 4 widely used declustering methods, a newly completed unified earthquake catalog for the Korean Peninsula and the adjacent regions and the assessment of its completeness. The electronic supplement is valuable too as it comprises: the homogeneous earthquake catalog, the declustered earthquake catalog based on each 4 methods and the completeness analysis results. I strongly recommend the article to be published, as important asset for the scientific community, with the mention that for a better view the figures 8, 9, 11, 13 to be a little more zoomed out.

**Reply:** Thank you for your positive feedback and for recommending our manuscript for publication. We also appreciate the reviewer's recognition of the importance of the unified earthquake catalog, the analysis of its completeness, and the inclusion of multiple declustering methods. As per your recommendation, we have updated the figures 8, 9, 11, 13 (Now Fig. 9, Fig. 10, Fig. 12, Fig. 14) in the revised manuscript to provide a better view. We appreciate your valuable suggestions and feedback.

**Response to Reviewers' Comments # 2 (RC2)**

We sincerely appreciate the reviewer thoughtful and constructive comments. The comments are all valuable and extremely helpful for revising and improving our manuscript, and they also provided important guidance for our research. We have carefully considered each comment and made revisions accordingly. We believe this study provides a necessary dataset and methodological framework that that will be useful for guiding and enhancing future seismic hazard assessments. Below, we provide a point-by-point response outlining the changes made in response to your comments.

Comment 1: The paper presents the compilation of a unified earthquake catalog, spanning 1905 to 2023, as well as a declustering analysis using four different methods and a completeness analysis of the compiled catalog. The paper is very well written and well presented. I have one significant comment and a few minor suggestions –

I will start from the end – in line 696 you write: This study provides valuable insights into seismic activities in the South Korean region and serves as a foundation for further research to optimize declustering methodologies for enhanced seismic risk evaluation and mitigation strategies." However, I feel that the insights from the work are very limited. What you present is a catalog compilation and a simple comparison between four methods of declustering. Relatively simple and not very inspiring. While you claim that "identifying the most effective method for removing dependent earthquakes is challenging because there is no inherently unique approach, and the elimination results are not absolute." - At the end when you (or someone else) goes to estimate seismic hazard – you will have to choose one of the catalogs. Which one would you choose and why? Will you run with all four? Simply running a comparison is not enough. It has been done before. You don't have to show a full PSHA here, but can you at least estimate a and b for a couple of different regions with your four declustering methods and comment about the significance of the difference? Maybe it doesn't make a difference after all? I think this type of analysis can take your paper one step forward and help determine whether or not the differences in the clustering approaches even matter

**Reply:** We sincerely appreciate the thoughtful and constructive feedback by the reviewer. The reviewer has rightly pointed out that while the compilation of a homogeneous earthquake catalog and comparison of declustering methods are an important first step, the study would benefit from further analysis to understand the practical implications of declustering choices on seismic hazard assessment. As suggested, we have now incorporated an additional analysis in the revised manuscript, where we estimate the Gutenberg-Richter parameters (a and b values) for the Homogeneous catalog and declustered earthquake catalog using each of the four algorithms. This analysis enables a clearer understanding of how each declustering method affects seismicity rate parameters and whether the differences are substantial enough to impact hazard estimations. We have added the detailed results and discussion of the analysis in Section 7 (Results and discussion) of the revised manuscript, along with Table 5 (For reference, the corresponding table is included below) summarizing the findings. The parameters are estimated for both the entire study area and the south korean mainland to more explicitly examine regional variations. This enhancement aligns with the reviewer's recommendation and helps demonstrate the practical implications of declustering choices beyond methodological comparison. The comparison of magnitude frequency distribution reveals that while b-values are relatively consistent across different declustering algorithms. However, notable variations were observed in the a-values, which reflect the overall rate of seismicity. These differences highlight the impact that declustering choices can have on seismicity rate estimates, a key input to PSHA. This initial analysis, based on spatially aggregated subregions, provides only a first-order insight. A more robust evaluation of the impact of declustering on seismic hazard would require detailed source-based studies, incorporating polygonal or tectonic source zones, which allow for refined source characterization and regional seismicity modeling. This will be the focus of a future study, where we will perform full PSHA computations based on thoroughly characterized of seismic sources.

The description is incorporated at Line 697 in the revised manuscript with tracked changes (Line 680 in Revised manuscript).

Table 5. Summary of results including Gutenberg–Richter a and b values for all declustered catalogs, along with the changes in a and b ( $\Delta a$ ,  $\Delta b$ ) relative to the homogeneous catalog.

| Method              | Entire Study Region |         |       |       | Mainland South Korea |         |       |                     |
|---------------------|---------------------|---------|-------|-------|----------------------|---------|-------|---------------------|
| Wiction             | a-value             | b-value | Δa    | Δb    | a-value              | b-value | Δa    | $\Delta \mathbf{b}$ |
| Homogeneous Catalog | 6.98                | 0.83    | -     | -     | 6.18                 | 1.26    | -     | -                   |
| Gardner Method      | 6.50                | 0.79    | -0.48 | -0.04 | 5.68                 | 1.15    | -0.50 | -0.11               |
| Uhrhammer Method    | 6.65                | 0.82    | -0.33 | -0.01 | 5.78                 | 1.16    | -0.40 | -0.10               |
| Reasenberg Method   | 6.72                | 0.8     | -0.26 | -0.03 | 5.87                 | 1.18    | -0.31 | -0.08               |
| Marsan Method       | 6.40                | 0.75    | -0.58 | -0.08 | 5.58                 | 1.11    | -0.60 | -0.15               |

Regarding the reviewer's important question about which catalog would ultimately be used for seismic hazard assessment - we acknowledge that this is a critical issue. However, we would like to respectfully clarify that the primary objective of the present manuscript is to establish a homogeneous earthquake catalog for the Korean Peninsula and to provide a systematic comparison of widely used declustering methodologies. The study is intended as a foundational contribution, aiming to highlight how different declustering techniques perform and how they affect the identification of mainshock sequences and event rates. Given the scope of this work, we have not designated a single declustering method as the "best" or most appropriate for hazard analysis. Instead, our intention is to use the insights gained from this comparison as a stepping stone toward more advanced modeling. These results can be applied in probabilistic seismic hazard assessments (PSHA) that incorporate detailed seismic source characterization (e.g., tectonic or polygonal source zones) to evaluate the impact of different declustering methods within that framework. This would enable assessment of the sensitivity of PSHA outcomes to the choice of declustering technique and help determine whether the differences are significant enough to justify selecting one catalog over others.

**Minor comments:**

Q 1. Section 2 - what are the boundaries of your region of interest (please state those explicitly, the reader does not need to infer them from figures). What are the distances between those boundaries to locations of interest within the peninsula? In other words - show you are not

cutting your region too small compared to the locations you want to compute seismic hazard at. For example - JeJu-Do is about 250 from the nearest boundary, Is that sufficient?

**Reply:** The spatial extent of the study region was already been explicitly stated in both the abstract and introduction sections of the manuscript (Line 78 in the revised manuscript with track changes). The catalog covers the geographic region bounded by latitudes 31° to 42° N and longitudes 122° to 132.5° E, ensuring comprehensive coverage of seismicity that could influence the Korean Peninsula. With regard to Jeju-do, while its southern boundary lies approximately 250 km from the edge of the study region, it is important to note that the majority of seismic threat to Jeju originates primarily from the southeastern portion of the Korean Peninsula, particularly near the East Sea, where the boundary extends over 500 km from Jeju. This configuration ensures sufficient buffer to capture relevant seismic sources affecting Jeju and the surrounding areas. In designing the catalog boundaries, we followed a standard approach in seismic hazard analysis by extending the region at least 250 km beyond the area of interest, particularly around major population centers and critical infrastructure. This boundary provides sufficient spatial coverage to include all relevant seismic sources influencing Korean peninsula, ensuring reliable hazard estimates.

**Q 2. Line 221 – the criteria for duplicate events is 70km and 30sec. This is quite arbitrary. What about consistency in magnitude estimations? Is that in the criteria? If you have a M2.0 and a M6.0 occurring at the same time within 70km, are those duplicate?**

Reply: In our study, the identification of duplicate events was primarily based on temporal (±30 seconds) and spatial (within 70 km) thresholds. In addition to these criteria, we also incorporated a magnitude consistency filter, where events were considered potential duplicates only if their reported magnitudes differed by less than ±0.1 units. All events that deviate from magnitude difference and satisfy the temporal and spatial initial criteria were then manually reviewed on a case-by-case basis to ensure accuracy and consistency in the final catalog. This thorough inspection allowed us to resolve ambiguities and select the most reliable entries. This combined approach of using time-distance criteria and manual checking has also been followed in other studies, such as Sawires et al. (2019), Grünthal and Wahlström (2012), and Wang et al. (2009). When duplicate events were identified, the final decision on which record to retain was based on a priority given to regional bulletins - with preference assigned first to the Korea Meteorological Administration

(KMA), followed by the Japan Meteorological Agency (JMA), and then the International Seismological Centre (ISC).

The information regarding magnitude criteria and manual review approach are added in the revised manuscript with track changes at Section 3.2, Line 238 (Line 229 in Revised manuscript) and the paragraph is modified accordingly to improve clarity.

**Q 3.** Line 255 – When you average magnitude estimations obtained using different approaches – please provide also the Standard deviation of the magnitude estimate. Not only the average. You can add that to your catalog.

**Reply:** Thank you for your valuable suggestion. We have included the standard deviation of the magnitude estimates in the homogeneous earthquake catalog, as requested. The updated catalog with their corresponding standard deviations, is uploaded in the Supplementary Material section. The information is also added in the revised manuscript with track changes at Line 291. (Line 280 in Revised manuscript).

**Q 4. Line 265 – It isn't clear which equation is used – 3 or 4? If only 4 – please do not present Eq3. If both – then what is the criteria?**

**Reply:** We acknowledge the confusion and have now clarified the methodology in the revised manuscript. Specifically, for the magnitude conversion from MJMA to MW, we have used both Equation 3 and Equation 4, and the average MW value from these two estimates was taken as the final converted magnitude. For the conversion from MD and MV magnitude to MW, we have used only Equation 4. The standard deviation associated with the magnitude estimates is now provided in the homogeneous catalog. The line is changed accordingly in the revised manuscript with track changes at Line number 299.

Q 5. Line 282 - The magnitude of an event cannot be different for vertical and horizontal components. Its a source parameter. It was derived in the original paper, but does not make sense here. If you proceed with the averaging - at least present Std of Mw estimate.

**Reply:** We agree that moment magnitude (Mw) is a source parameter and should be independent of recording component. However, in the referenced publication, two separate empirical relationships were provided - one based on horizontal components and another on the vertical component. In our study, we used both relationships as published and estimated the average Mw

to ensure consistency. To address the concern, the standard deviation of the MW estimates are now included in the homogeneous earthquake catalog.

**Q** 6. Line 345 – I think there is a typo in the distance equation. Why would 1.77 be outside the brackets and 0.037 inside if both are constants?

**Reply:** This was a typographical error. We have now corrected Equation 10 in the revised manuscript accordingly. (Line 381 in revised manuscript with track changes).

Q 7. Line 438 – "with particular attention paid to the temporal smoothness of the resulting catalog" – what does that mean? what is temporal smoothness?

**Reply:** To improve clarity, we have removed the term "temporal smoothness", as we recognize it may be ambiguous. We have revised the sentence to avoid this term and better reflect the actual practice. The revised sentence now reads:

"Depending on the declustering results, a trial-and-error process is often used, with particular attention to how different parameter choices influence the separation between background and clustered events, which may vary across studies and introduce inconsistencies."

The revised sentence is incorporated at Line 477 in the revised manuscript with track changes. (Line 462 in Revised manuscript).

Q 8. Line 466 – I think there is a typo in this sentence in using "of which". This reads as if you identified 25,229 mainshocks, and 38,069 out of the 25,229 were identified as aftershocks. This does not make sense.

**Reply:** This was a typo error, and we have revised the sentence accordingly in the revised manuscript. The corrected sentence now reads:

"From a homogeneous dataset of 63,298 events, the MISD algorithm identified 25,229 mainshock events and 38,069 aftershocks."

The correct sentence is incorporated at Line 508 in the revised manuscript with track changes. (Line 491 in Revised manuscript).

**Q 9. Line 511** – This is still the main portion of the text. Give section number instead.

Reply: Instead of introducing a new section, we have revised the text to explicitly refer to the

current section number. The phrase "main portion of the text" has been replaced with "discussed

subsequently," indicating that it is addressed within the current section. Additionally, we have

compared the Gardner catalog results and the results from the other declustered catalogs within

distinct paragraphs in the same section. This approach ensures a clear and coherent flow while

keeping the analysis within the same section, as it is a comparative analysis. Detailed plots of the

completeness analysis for the alternative declustering methods are provided in the electronic

supplementary material (Completeness analysis.docx), as now mentioned in the revised

manuscript with track changes at Line 555. (Line 537 in Revised manuscript).

Q 10. Line 540 – 'bins versus time' – which time? What is 10, and 100? Is that "time before

present"? If so – then it should say so in the text and figure caption.

**Reply:** In this context, time represents the length of the time window, expressed in years, used for

calculating the standard deviation of the mean annual earthquake rate. The time span is 10 years

used in the present study, starting from the present year (2023) and extending backward into the

past. This information was not explicitly stated in the manuscript and has now been added in revised

manuscript to the section discussing the Stepp (1972) approach as well as in the corresponding

figure caption (Line 585 in the revised manuscript with track changes). Additionally, the x-axis

title in Figure 15 has been updated from "Time" to "Time Interval" for clarity.

**Q 11. Section 7** – The comparison results from both declustering and completeness analysis

should be compared with similar studies in the literature.

e.g.

DOI: 10.1007/s10950-024-10221-8

DOI: 10.1785/0220210127.

And others...

**Reply:** We have included a comparative discussion of our declustering results with those reported

in similar studies, including the works cited by the reviewer. Specifically, we have prepared a

summary table comparing the percentage of events removed and the proportion of identified

mainshocks for each declustering algorithm applied in our study, and contrasted these results with

those from other regional studies worldwide. While we understand that these comparison studies

focus on different tectonic settings, the methodological similarities allow for a meaningful comparison of the relative performance and behavior of the declustering techniques. This additional comparison has been incorporated into the revised manuscript to enhance the contextual understanding of our findings. The comparison Table 4 is updated and a new comparison table summarizing the results is now included in the revised manuscript. The Table 4 is also attached here for reference.

The description and updated table is incorporated in the revised manuscript with track changes at Line 681 (Line 667 in the Revised manuscript).

Table 4. Declustering Results for the Korean Peninsula and Summary of Findings from Selected Studies in Other Regions.

| Study                    | Decluster Method   | Mainshocks  | Aftershocks | Notes                            |
|--------------------------|---------------------------|-------------|-------------|----------------------------------|
| Present Study            | Gardner Method            | 30912 (49%) | 32386 (51%) |                                  |
|                          | Uhrhammer Method          | 38572 (61%) | 24726 (39%) | — Region: Korean — Peninsula and |
|                          | Reasenberg Method         | 39978 (63%) | 23320 (37%) | — Peninsula and surrounding      |
|                          | Marsan Method             | 25229 (40%) | 38069 (60%) |                                  |
| Perry and Bendick (2024) | GK                        | 3018 (19%)  | 12876 (81%) |                                  |
|                          | Reasenberg                | 2855 (18%)  | 13039 (82%) | — Ionan (2010                    |
|                          | Uhrhammer                 | 4410 (28%)  | 11484 (72%) | — Japan (2010-
— 2018)        |
|                          | Zhuang-ETAS
Stochastic | 6001 (38%)  | 9893 (62%)  | 2010)                            |
| Nas et.al (2019)         | GK                        | 6713 (51%)  | 6593 (49%)  |                                  |
|                          | Reasenberg                | 11420 (85%) | 1886 (15%)  | Turkey catalog                   |
|                          | Uhrhammer                 | 9009 (67%)  | 4297 (33%)  | (1900 -2016)                     |
|                          | ETAS                      | 6959 (52%)  | 6347 (48%)  |                                  |
| Poudyal et. al (2025)    | GK                        | 1466 (45%)  | 1724 (54%)  | — Kathmandu                      |
|                          | Reasenberg                | 2313 (72%)  | 877 (28%)   | — Valley                         |
|                          | Uhrhammer                 | 1770 (55%)  | 1420 (45%)  | — vaney                          |
| Perry and Bendick (2024) | GK                        | 2891 (44%)  | 3633 (56%)  |                                  |
|                          | Reasenberg                | 5222 (80%)  | 1302 (20%)  | Northern                         |
|                          | Uhrhammer                 | 4539 (70%)  | 1985 (30%)  | Rockies                          |
|                          | Zhuang-ETAS
Stochastic | 1862 (29%)  | 4662 (71%)  | (Canada)                         |

\* GK = Gardner and Knopoff (1974); ETAS = (Zhuang et al., 2002).

Q 12. Lines 660 until end of paragraph - There is nothing new stated in this paragraph. There are always more small to moderate earthquakes than large ones. Everywhere. The G-R model

shows that, as do others. If you only have reliable data from the past 40 years you probably need models to help you estimate the large events. The text here is trivial.

**Reply:** We agree that the predominance of small-to-moderate earthquakes than large one is a general characteristic of global seismicity. Our intention in this paragraph was not to present this as a finding, but rather to contextualize the seismic behavior of the Korean Peninsula using a carefully prepared homogeneous and declustered earthquake catalog. By reducing biases associated with catalog inconsistencies and aftershock sequences, we aimed to provide a clearer view of background seismicity in the region. We have revised the paragraph and avoid redundancy. We have revised the paragraph as:

"The seismic activity pattern in the region reveals that no large ( $Mw \ge 6$ ) earthquakes have occurred in the Korean Peninsula over the past 40 years, with earthquake occurring predominantly in the  $M_W$  2-5 range. This implies that the region predominantly experienced low-to-moderate seismic activity. The earthquake cluster indicates a relatively stable seismic environment for the Korean Peninsula, with occasional moderately high tremors. While this might suggest a relatively low level of seismic hazard based on recent activity alone, historical records and paleoseismic studies indicate that larger, potentially damaging earthquakes have occurred in the region. Thus, the absence of large earthquakes in recent decades should not be interpreted as an assurance of long-term stability. Therefore, although recent data indicate a reduced likelihood of significant seismic events, the limitations of earlier data introduce a degree of uncertainty in long-term seismic hazard assessments. This highlights the importance of considering both instrumental and historical information when evaluating the regional seismic hazard and risk analysis."

The revised the paragraph is added in Line 765 in the revised manuscript with track changes (Line 749 in Revised manuscript).

Q 13. Figure 18 – It is quite hard to gain anything from these figures, and I do not believe you say anything significant about them in the text. Are they really necessary? What can you say about the geographical distribution that can only be understood from these plots (and not the maps for example)?

**Reply:** The maps included in the manuscript illustrate the geographical distribution of earthquakes (latitude and longitude), while the 3D plot was intended to emphasize the temporal distribution of seismicity over the past decades. Specifically, this 3D plot allows for the visualization of how

seismicity has evolved over time in both space and magnitude. By including time as an axis and representing magnitude through color, the 3D plot helps to visualize the absence of large or potentially damaging earthquakes in the recent 40-year instrumental period, compared with the occurrence of larger events in earlier decades. This pattern is more difficult to interpret from 2D maps, which often depicts the spatial coverage. The 3D visualization enables a holistic view of how seismicity is distributed across space and time simultaneously, helping to support our discussion of the limitations of recent data for long-term seismic hazard assessment.

**Response to Reviewers' Comments # 3 (RC3)**

We sincerely thank the reviewer for the constructive comments and helpful suggestions, which have significantly improved the clarity and quality of the manuscript. In response, we have carefully revised the text and updated the relevant figures to enhance the overall presentation and ensure clearer understanding of the results. We hope the revised manuscript meets the expectations and is now suitable for publication.

**Comment Reviewer #3:**

The study compiles an extensive earthquake catalog (1905–2023) using multiple reliable sources (KMA, ISC, JMA) and ensures consistency through magnitude conversion. It offers a robust seismicity assessment by applying four declustering methods and using Stepp's method and CUVI for completeness analysis. The research highlights earthquake clustering in South-eastern Korea, aiding seismic hazard and risk assessments, and provides a strong foundation for microzonation and engineering applications. Its well-structured methodology enhances clarity and reproducibility. However, certain aspects require further point-by-point explanations:

**Q1.** More recent literature on earthquake catalogs of the Korean Peninsula could be added to the introduction section to clarify how this study stands out in comparison.

**Reply:** Thank you for your valuable comment. We would like to clarify that a literature review discussing previous catalog works in Korea were already been included in the original manuscript. Please refer to Line 51 in the revised manuscript with track changes (Line 50 in Revised manuscript). Our study aimed to prepare a homogeneous earthquake catalog encompassing the entire Korean Peninsula, which distinguishes it from earlier efforts that were either region-specific or limited in scope or methodology. In addition to the detailed descriptions and the updated catalog have been provided as electronic supplementary material. This is intended to support a better understanding of seismic activity in the region and contribute to future research in seismic hazard. For ease of reference, the relevant sentences from the manuscript are again provided below:

"Studies on earthquake catalogs in Korea have been conducted over several decades, with significant contributions from Li (1986), Kim and Gao (1995), and Lee (1999). Since the Korea

Meteorological Administration (KMA) has strengthened its national seismological observation network, recent efforts have focused primarily on estimating historical earthquakes (Lee & Yang, 2006; Seo et al., 2010). Seismic hazard studies in Korea typically use earthquake data from the KMA database (Han & Choi, 2008; Kyung et al., 2016). Ideally, a comprehensive earthquake catalog should be compiled by integrating earthquake data from all available sources, not just regional ones. Recent seismic hazard research by Park et al. (2021) identified this issue and incorporated instrumental earthquake catalogs from the KMA, JMA, and the China Earthquake Administration (CEA) for their analysis. However, their database was limited to South Korea, and their primary focus was on seismic hazard studies rather than catalog details. By contrast, our study aimed to prepare a homogeneous catalog encompassing the entire Korean Peninsula. In addition, detailed descriptions and an updated catalog are provided as electronic supplementary material, intended to aid in understanding seismic activity in the region and to enhance earthquake-related research and preparedness efforts."

**Q2.** Figure 3 and 4, M represents various types of magnitude scales. If the magnitudes scales could be defined by separate colours it would be better.**

**Reply:** In accordance with the comment, Figures 3 and 4 have been modified in the revised manuscript. Different types of magnitude scales are now represented using distinct colors to improve clarity and visual interpretation. The modified figures are included in the revised manuscript with track changes at Line 185 and Line 209. For the convenience, the modified figures are also provided below.

Figure 3: Seismicity distribution of earthquake locations from the International Seismological Centre (ISC) bulletin. In this figure, different magnitude scales, including  $M_b$ ,  $M_{JMA}$ ,  $M_s$ ,  $M_W$ ,  $M_L$ ,  $M_D$  and  $M_V$  are represented using distinct colors.

Figure 4: Seismicity distribution of earthquake locations from the Japan Meteorological Agency (JMA) source. In this figure, different magnitude scales, including  $M_b$ ,  $M_{JMA}$ ,  $M_D$  and  $M_V$  are represented using distinct colors.

Q3. Annual reporting of earthquakes with magnitude (all types) ≥2.0 in the study region from the three major agencies: ISC, KMA and JMA database can be shown in a comparative plot for better understanding of the event counts and temporal variation.

**Reply:** We have plotted the annual earthquake counts ( $M \ge 2.0$ ) for the three agencies namely KMA, JMA, and ISC in a comparative plot, allowing direct visualization of the temporal variation and differences in reported events across agencies. The description and figure (Fig. 5) are added to the revised manuscript with track changes in Section 3.2 at Line 220 (Line 211 in Revised manuscript). The figure is also provided here for reference purposes.

**Annual Earthquake Counts by Agency**

Figure 5: Annual earthquake counts reported by KMA, JMA, and ISC in the Korean Peninsula. The plot allows comparison of temporal variations in seismic reporting among the three agencies.

**Q4.** The study uses several magnitude conversion equations from past research. Were any validations performed on the converted magnitudes to check for biases or inconsistencies?

**Reply:** We would like to clarify that the present study did not derive or apply any new magnitude conversion equations. Instead, we have utilized well-established and widely accepted global conversion equations that were specifically developed for the regions or magnitude scales relevant to this study. The validation and evaluation of these equations were thoroughly performed by the original authors in their respective publications. Our study has adopted these published equations as they are, considering their established reliability and recognition within the seismological community.

**Q 5.** This study shows a comparison between four different techniques for declustering of the events comprehensively. What do you think? Which one is the best in this case?**

**Reply:** We sincerely thank the reviewer for raising this thoughtful question regarding the optimal declustering method and its implications for seismic hazard assessment. We fully acknowledge that the choice of declustering technique can significantly influence seismic hazard modeling outcomes. While the primary objective of this manuscript is to contribute to the development of a homogeneous earthquake catalog for the Korean Peninsula and present a systematic comparison of widely used declustering methods, we have included additional analysis to evaluate the practical implications of the declustering choices.

Specifically, we now compare the Gutenberg-Richter parameters (a- and b-values) derived from each of the declustered catalogs, alongside the homogeneous catalog, as a way to assess how the different methods impact seismicity rate estimates. This analysis, will be included in the revised manuscript at Section 7 (Table 5), which will provide insight into the degree to which declustering affects frequency-magnitude distributions. The description is incorporated at Line 697 in the revised manuscript with track changes (Line 680 in Revised manuscript). While this analysis provides valuable insight into the sensitivity of seismicity rate estimates to different declustering approaches, we refrain from selecting a single "best" method at this stage. A full evaluation of how these declustering methods affect seismic hazard estimation especially within a Probabilistic Seismic Hazard Assessment (PSHA) framework requires further dedicated analysis involving detailed seismic source modeling and hazard computations, which extends beyond the scope of the present study. We plan to address this critical aspect in a future study, where full PSHA computations will be performed using the different declustered catalogs to assess their influence on hazard results.

Q 6. The study applies four different declustering techniques to identify mainshocks and remove dependent events. Were foreshocks considered in the declustering process, or does the analysis focus only on aftershocks? If not, could these methods be adapted to distinguish foreshocks as well?

**Reply:** We would like to clarify that the declustering algorithms applied in this study are commonly employed to distinguish mainshocks from dependent events, which include both aftershocks and foreshocks. All four methods employed including window-based, cluster-based and stochastic based treat dependent events comprehensively and are not limited to aftershocks alone. This aspect is already mentioned in the original submitted manuscript (please refer to lines 364 and 424 in the revised manuscript with track changes), where we describe the scope of the declustering techniques. As such, the analysis does not exclude foreshocks; rather, it systematically identifies and removes all dependent events, including both aftershocks and foreshocks, from the mainshock catalog.